# Stochastic Matching Bandits under Preference Feedback

## Abstract

In this study, we propose a new bandit framework of stochastic matching employing the Multinomial Logit (MNL) choice model with feature information. In this framework, agents on one side are assigned to arms on the other side, and each arm stochastically accepts an agent among the assigned pool of agents based on its unknown preference, allowing a possible outside option of not accepting any. The objective is to minimize regret by maximizing the probability of successful matching. For this framework, we first propose an elimination-based algorithm that achieves a regret bound of $\widetilde{\mathcal{O}}\big(K\sqrt{rKT}\big)$ over time horizon $T$, where $K$ is the number of arms and $r$ is the rank of feature space. Furthermore, we propose an approach to resolve the computation issue regarding combinatorial optimization in the algorithm. Lastly, we evaluate the performances of our algorithm through experiments comparing with the existing showing the superior performances of our algorithm.

## 1 Introduction

In recent times, the rising prevalence of matching markets, such as ride-hailing services, online job markets, and online labor markets, has increased the demand for strategies to enhance successful matching. In ride-hailing services, for instance, the goal is to pair riders with drivers to maximize the probability of successful matched rides, thereby increasing revenue. In online job markets, the goal is to maximize successful employment between applicants and employers.

Due to the demand for matching problems, online (bipartite) matching problems have been widely studied (Karp et al., 1990; Mehta et al., 2007; 2013; Gamlath et al., 2019; Fuchs et al., 2005; Kesselheim et al., 2013). In these problems, there are two sides for vertices where vertices are revealed one at a time. Then the focus is on optimizing to maximize matching. However, there exists a gap between the models considered in online matching problems and their real-world applications. For ease of presentation, in what follows, we refer to the vertices on one side as agents and those on the other side as arms. Previous studies only focused on assigning a single agent to an arm, not allowing the assignment of multiple agents to an arm. More crucially, based on these setups, the focus has been solely on optimization to maximize matching rather than learning underlying models from bandit feedback. In real-world applications such as ride-hailing services, however, preferences are unknown in advance, and learning the latent preferences of drivers is necessary while maximizing the probability of call acceptance.

Based on the demand for online learning in real-world applications, sequential matching under bandit feedback, referred to as *matching bandits*, has recently gained attention (Liu et al., 2020; 2021; Sankararaman et al., 2020; Basu et al., 2021; Zhang et al., 2022; Kong & Li, 2023). The typical problem setting studied in the matching bandits is as follows. In each round, each agent is assigned to an arm, and then the arms accept an agent among the assigned pool of agents, which results in the completed matching(s) of an agent and an arm. Then, stochastic reward feedback corresponding to each match is observed, where the true distributions of rewards are assumed to be unknown. The objective is to find stable matching based on the preferences as in the stable marriage problems McVitie & Wilson (1971) by learning reward distributions.

However, there is still a gap between models considered in the matching bandit literature and real-world applications. The previous studies presume that the behavior of arms is *deterministic* based on

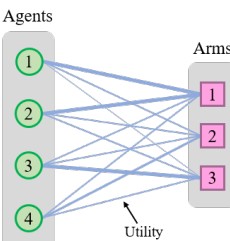

Agents

Arms

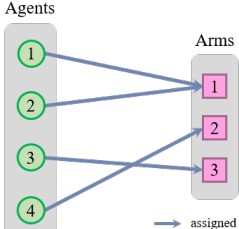
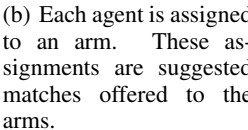

Agents

Arms

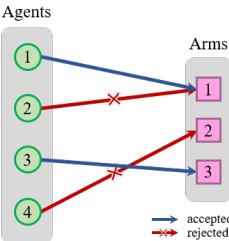

Agents

Arms

(a) There exist different values of utility between agents and arms. The true utility values are unknown.

(b) Each agent is assigned to an arm. These assignments are suggested matches offered to the arms.

(c) Each arm accepts one of the suggested matches or rejects all of them.

Figure 1: Illustration of our stochastic matching process with 4 agents ($N = 4$) and 3 arms ($K = 3$).

known preferences without allowing refusal from arms. That is, given the assignments of agents to arms, the matching outcome is deterministically given by the preferences that each arm has. Based on the deterministic behavior, the previous work focuses on finding stable matching. However, this assumption may not align with many practical applications, as seen, for instance, in ride-hailing services, where there are riders (agents) and drivers (arms). When the dispatch system assigns riders' calls to drivers, the choice behavior of drivers for accepting calls from riders is *stochastic* rather than deterministic. This stochastic acceptance is not just among multiple calls assigned to a single driver, but also for single assigned call. That is, the driver may refuse a call according to some choice preference, which is unknown to riders or even to the dispatch system. Then, a critical question arises:

*How can we maximize the success of matching under this stochastic behavior?*

In this work, we introduce a novel and practical online matching framework, termed *stochastic matching bandits*, to describe the bandits for matching operating under stochastic matching conditions characterized by unknown preferences. It is important to highlight that our framework is fundamentally different from existing approaches, such as online matching problems and matching bandits. We describe our proposed framework with an illustration in Figure 1 as follows. **(a)** Multiple agents and arms are involved with unknown utility values between them reflecting the arms' preferences over agents. **(b)** In each time step, each agent is assigned to an arm. Note that multiple agents can be assigned to a single arm. These matching assignments are proposed to the arms, but the final matching has not yet been confirmed. **(c)** Arms accept a suggested matching from an agent among the assigned pool of agents based on their preferences or reject all of them, making stochastic decisions under unknown preferences.

For the choice model of arms over agents, we use the multinomial logit (MNL) function with features. Significantly, our model incorporates outside options, allowing arms to opt out of accepting any agents from the assigned pool. This consideration of outside options reflects many real-world scenarios, such as drivers in ride-hailing services choosing not to accept calls from assigned riders based on their preferences, or employers in online job markets deciding not to hire any. We highlight that, in this problem, the success or failure of the match serves as the reward feedback for agents. Then, our objective is to maximize the probability of successful matching in the systems while learning unknown preferences in an online manner.

**Summary of Our Contributions.** In the following, we provide a summary of our contributions.

- We introduce a novel framework of stochastic matching bandits, which incorporates the stochastic behavior of arms along with the inclusion of outside options (refusals of assigned matching). This framework entails the reward of success or failure from stochastic matching under latent preferences given by a MNL model. Moreover, our feature-based MNL modeling in the proposed framework allows for generalization and efficient learning across agents.

- Under our stochastic matching bandits framework, we propose an elimination-based algorithm SMB that efficiently balances exploration and exploitation.

- We establish the regret bound of our proposed algorithm, achieving $\widetilde{\mathcal{O}}\big(K\sqrt{rKT}\big)$ regret, where $r$ is the rank of feature space, $K$ is the number of arms, and $T$ is the time horizon.

- Furthermore, we explore a method to address the computation issue regarding combinatorial optimization for the algorithm.

- Finally, we demonstrate the numerical outperformances of our algorithm in comparison with previously proposed matching bandit algorithms.

## 2 RELATED WORK

**Online Matching Problem.** In regards to matching optimization, the online matching problem has been studied by Karp et al. (1990); Mehta et al. (2013; 2007); Gamlath et al. (2019); Fuchs et al. (2005); Kesselheim et al. (2013), in which the objective is to maximize matching from irreversible matching decisions under real-time arrivals. Online bipartite matching was first studied by Karp et al. (1990), which provided a randomized algorithm for the online matching problem in a bipartite graph. Subsequent research has expanded upon this foundation, with studies such as Kesselheim et al. (2013) addressing weighted bipartite matching and Gamlath et al. (2019) exploring the generalized arrival model regarding vertices and edges. In contrast to the typical focus on optimization in online matching problems, our study, inspired by bandit research, concentrates on learning preference utilities related to preferences by allowing the assignment of multiple agents to an arm. Moreover, for the utility values, we explore a more generalized structure with features, which has not been studied for the online matching problem. Importantly, our approach introduces a fundamentally different aspect from the online matching problems by addressing the tradeoff between exploitation and exploration.

**Matching Bandits.** Here we examine the existing body of literature on matching bandits. The exploration of regret minimization in matching markets was initially studied by Liu et al. (2020). Their objective was to minimize regret by finding an optimal stable matching by learning agents' side preferences through stochastic reward feedback. In their work, the Explore-then-Commit-based algorithm, integrated with the Gale-Shapley platform (Gale & Shapley, 1962), was proposed. Subsequent studies by Sankararaman et al. (2020); Liu et al. (2021); Basu et al. (2021) focused on decentralized agents within matching bandits. More recently, Zhang et al. (2022); Kong & Li (2023) accomplished improved regret bounds. The previous work does not consider feature information or general structure, focusing solely on the basic MAB setting. We also note that all of the previous work only considered $N \leq K$.

Our research distinguishes itself from prior work on matching bandits in several aspects. Firstly, prior work focuses on achieving a stable matching under the assumption that arm behavior is deterministic with known preferences. This implies that each arm accepts an agent deterministically based on its known preference, given the assigned agents without outside options (that is, there is no option to not choose any agent). In contrast, we posit that arm behavior is stochastic with unknown preferences, highlighting the importance of learning arms' preferences. Furthermore, we allow arms to have outside options (not to accept any agent). Thus, successful matching by avoiding the outside option is crucial. *This naturally directs focus towards maximizing the likelihood of successful matches while acquiring preference feedback rather than focusing on stability*. This indicates that our setting does not align with previous research but instead represents a distinct problem. Additionally, we consider preference utility as a function of features. Lastly, we do not assume any relationship between $N$ and $K$, allowing for $N \geq K$ or $N \leq K$.

**MNL-Bandits.** As the first MNL bandit method, Agrawal et al. (2019) proposed an epoch-based algorithm, followed by subsequent contributions from Agrawal et al. (2017b); Chen et al. (2023); Oh & Iyengar (2019; 2021). In our study, we adopt the MNL model for arms' choice preferences in matching bandits, which has not been studied before. Unlike selecting an assortment at each time step, our novel framework for the stochastic matching market mandates choosing at most $K$ distinct assortments to assign agents to each arm.

## 3 PROBLEM STATEMENT

There are $N$ agents and $K$ arms. For each agent $n \in [N]$, feature information is known as $x_n \in \mathbb{R}^d$, and each arm $k \in [K]$ is characterized by latent vector $\theta_k \in \mathbb{R}^d$. We define the set of features as $X = [x_1, \ldots, x_N] \in \mathbb{R}^{d \times N}$ and the rank of $X$ as $rank(X) = r(\leq d)$. At each time $t \in [T]$, every agent $n$ may be assigned to an arm $k_{n,t} \in [K]$. Let assortment $S_{k,t} = \{n \in [N] : k_{n,t} = k\}$, which is the set of agents that are assigned to an arm $k$ at time $t$. We consider that $|S_{k,t}| \leq L$ for all $k$ and $t$. Then given an assortment to each arm $k$ at time $t$, $S_{k,t}$, each arm $k$ randomly accepts an agent $n \in S_{k,t}$ according to the arm's preference specified as follows. The probability for arm $k$ to accept agent $n \in S_{k,t}$ follows Multi-nomial Logit (MNL) model (Agrawal et al., 2017a;b; Oh & Iyengar, 2019; 2021; Chen et al., 2023) given by

$$p(n|S_{k,t}, \theta_k) = \frac{\exp(x_n^\top \theta_k)}{1 + \sum_{m \in S_{k,t}} \exp(x_m^\top \theta_k)}.$$

We note that $x_n^\top \theta_k$ represents the preference utility of arm $k$ to agent $n$. Then at each time $t$, the agents observe the stochastic matching feedback of the assortments $\{S_{k,t}\}_{k \in [K]}$ as $y_{n,t} \in \{0, 1\}$ for all $n \in S_{k,t}$, $k \in [K]$, in which $y_{n,t} = 1$ when arm $k$ accepts agent $n$ (arm $k$ is matched with agent $n$), and otherwise $y_{n,t} = 0$. Importantly, as considered in the previous MNL bandit literature, it is allowed to have an outside option ($n_0$) for each arm $k$ according to MNL with probability as

$$p(n_0|S_{k,t}, \theta_k) = \frac{1}{1 + \sum_{m \in S_{k,t}} \exp(x_m^\top \theta_k)}.$$

We define the probability that arm $k$ is matched with any agent in $S_k$ as $R_k(S_k) = \sum_{n \in S_k} p(n|S_k, \theta_k)$. Then, given assortments to every arm $k$, $\{S_k\}_{k \in [K]}$, the expected number of successful matches at time $t$ is

$$\sum_{k \in [K]} R_k(S_k) = \sum_{k \in [K]} \sum_{n \in S_k} p(n|S_{k,t}, \theta_k) = \sum_{k \in [K]} \sum_{n \in S_k} \frac{\exp(x_n^\top \theta_k)}{1 + \sum_{m \in S_k} \exp(x_m^\top \theta_k)}.$$

The purpose of this problem is to maximize the expected number of successful matches over horizon time $T$ while learning latent $\theta_k$'s. Here we define the oracle strategy, which is the optimal policy when $\theta_k$'s are known. Define the set of all possible assortments to be $\mathcal{M} = \{\{S_k\}_{k \in [K]} : S_k \subset [N], |S_k| \leq L \, \forall k \in [K], S_k \cap S_l = \varnothing \, \forall k \neq l\}$. Then the oracle strategy is the following: $\{S_k^*\}_{k \in [K]} = \arg\max_{\{S_k\}_{k \in [K]} \in \mathcal{M}} \sum_{k \in [K]} R_k(S_k)$. Given $\{S_{k,t}\}_{k \in [K]} \in \mathcal{M}$ for all $t \in [T]$, the expected regret is defined as

$$\mathcal{R}(T) = \mathbb{E}\left[\sum_{t \in [T]} \sum_{k \in [K]} R_k(S_k^*) - R_k(S_{k,t})\right].$$

Then the goal of this problem is to find a policy to minimize the regret. Next, we provide assumptions for some regularity conditions in this problem as follows.

**Assumption 1.** $\|x_n\|_2 \leq 1$ for all $n \in [N]$ and $\|\theta_k\|_2 \leq 1$ for all $k \in [K]$.

**Assumption 2.** There exists $\kappa > 0$ such that for any $n \in S$ and $S \subset [N]$, $\inf_{\theta \in \mathbb{R}^d : \|\theta\|_2 \leq 2} p_k(n|S, \theta) p_k(n_0|S, \theta) \geq \kappa$ for all $k \in [K]$.

It is worth mentioning that these regularity conditions are commonly taken into account in the logistic and MNL bandit literature (Faury et al., 2020; Abeille et al., 2021; Oh & Iyengar, 2019; 2021).

**Notations.** Let $[a, b]$ denote the set of integers from $a$ to $b$. For a vector $x \in \mathbb{R}^d$ and a positive definite matrix $A \in \mathbb{R}^{d \times d}$, $A$-weighted norm of $x$ is denoted by $\|x\|_A = \sqrt{x^\top A x}$.

## 4 ALGORITHM AND REGRET ANALYSIS

To handle the large number of possible matchings between agents and arms while learning preferences, we propose an elimination-based algorithm (Algorithm 1), taking insights from Lattimore &

---

**Algorithm 1** Stochastic Matching Bandit (SMB)

---

**Input:** $T, \kappa, C_1 > 0, C_2 > 0, C_3 > 0$

**Init:** $t \leftarrow 1, T_1 \leftarrow C_2 \log(KNT), \mathcal{M}_0 \leftarrow \mathcal{M}, \mathcal{N}_{k,0} \leftarrow [N], \mathcal{T}_{n,k,0}^{(2)} = \varnothing$ for all $k \in [K], n \in [N]$

1: Compute SVD of $X = U\Sigma V^\top$ and obtain $U_r = [u_1, \ldots, u_r]$; Construct $z_n \leftarrow U_r^\top x_n$ for $n \in [N]$

2: Run `Round-robin Warm-up` (Algorithm 2) over time steps in $\mathcal{T}_k^{(1)}$ (defined in Algorithm 2) for $k \in [K]$

3: **for** $\tau = 1, 2 \ldots$ **do**

4:   **for** $k \in [K]$ **do**

     // Estimation

5:     $\mathcal{T}_{k,\tau-1} \leftarrow \mathcal{T}_k^{(1)} \cup \mathcal{T}_{k,\tau-1}^{(2)}$ where $\mathcal{T}_{k,\tau-1}^{(2)} := \bigcup_{\tau' \in [\tau-1]} \bigcup_{n \in \mathcal{N}_{k,\tau'}} \mathcal{T}_{n,k,\tau'}^{(2)}$

6:     $V_{k,\tau-1} \leftarrow \sum_{s \in \mathcal{T}_{k,\tau-1}} \sum_{n \in S_{k,s}} z_n z_n^\top$

7:     $\hat{\theta}_{k,\tau-1} \leftarrow \arg\min_{\theta \in \mathbb{R}^r} l_{k,\tau-1}(\theta)$ with (1).

     // Assortments Construction

8:     $\{S_{l,\tau}^{(n,k)}\}_{l \in [K]}$

     $\leftarrow \arg\max_{\{S_l\}_{l \in [K]} \in \mathcal{M}_{\tau-1}: n \in S_k} \sum_{l \in [K]} R_{l,\tau-1}^{UCB}(S_l)$ for all $n \in \mathcal{N}_{k,\tau-1}$ with (2)

     // Elimination

9:     $\mathcal{N}_{k,\tau} \leftarrow \{n \in \mathcal{N}_{k,\tau-1} : \max_{\{S_l\}_{l \in [K]} \in \mathcal{M}_{\tau-1}} \sum_{l \in [K]} R_{l,\tau-1}^{LCB}(S_l) \leq \sum_{l \in [K]} R_{l,\tau-1}^{UCB}(S_{l,\tau}^{(n,k)})\}$

    with (2)

     // G/D-optimal design

10:    $\pi_{k,\tau} \leftarrow \arg\max_{\pi \in \mathcal{P}(\mathcal{N}_{k,\tau})} \log \det \sum_{n \in \mathcal{N}_{k,\tau}} \pi_{k,\tau}(n) z_n z_n^\top$

     // Exploration

11:    **for** $n \in \mathcal{N}_{k,\tau}$ **do**

12:      $t_{n,k} \leftarrow t, \mathcal{T}_{n,k,\tau}^{(2)} \leftarrow [t_{n,k}, t_{n,k} + \lceil r\pi_{k,\tau}(n)T_\tau \rceil - 1]$

13:      **while** $t \in \mathcal{T}_{n,k,\tau}^{(2)}$ **do**

14:        Offer $\{S_{l,t}\}_{l \in [K]} = \{S_{l,\tau}^{(n,k)}\}_{l \in [K]}$

15:        Observe feedback $y_{m,t} \in \{0, 1\}$ for all $m \in S_{l,t}$ and $l \in [K]$

16:        $t \leftarrow t + 1$

17:   $\mathcal{M}_\tau \leftarrow \{\{S_k\}_{k \in [K]} : S_k \subset \mathcal{N}_{k,\tau}, |S_k| \leq L \,\forall k \in [K], S_k \cap S_l = \varnothing \,\forall k \neq l\}$

18:   $T_{\tau+1} \leftarrow 2T_\tau$

---

Szepesvári (2020) and Chen et al. (2023). The main idea of this algorithm is to eliminate feasible connections between agents and arms delicately that are identified as suboptimal over epochs. The details are described in the following.

Before advancing on the rounds, the algorithm computes Singular Value Decomposition (SVD) for feature matrix $X = U\Sigma V^\top \in \mathbb{R}^{d \times N}$. From $U = [u_1, \ldots, u_d] \in \mathbb{R}^{d \times d}$ and $rank(X) = r$, we can construct $U_r = [u_1, \ldots, u_r] \in \mathbb{R}^{d \times r}$ by extracting the left singular vectors from $U$ that correspond to non-zero singular values. We note that the algorithm does not necessitate prior knowledge of $r$ because the value can be obtained from SVD. The algorithm, then, operates within the full-rank $r$-dimensional feature space with $z_n = U_r^\top x_n \in \mathbb{R}^r$ for $n \in [N]$. The insight behind this approach is as follows.

Since $x_n$ for $n \in [N]$ lies in the subspace $U_r$, we observe that $x_n = U_r b_n$ for some $b_n \in \mathbb{R}^r$. Let $\theta_k^* = U_r^\top \theta_k$. Then we have $x_n^\top \theta_k = z_n^\top \theta_k^*$ by following $x_n^\top \theta_k = b_n^\top U_r^\top \theta_k = b_n^\top (U_r^\top U_r) U_r^\top \theta_k = x_n^\top U_r U_r^\top \theta_k = z_n^\top \theta_k^*$ using $U_r^\top U_r = I_d$. Therefore, we can reformulate the MNL model using $r$-dimensional feature $z_n \in \mathbb{R}^r$ and latent $\theta_k^* \in \mathbb{R}^r$ in place of $d$-dimensional $x_n \in \mathbb{R}^d$ and $\theta_k \in \mathbb{R}^d$, respectively, for $n \in [N]$ and $k \in [K]$. We note that this procedure is beneficial not only for reducing feature dimension but also for introducing appropriate regularization for estimators without imposing any assumption about feature distributions considered in Oh & Iyengar (2021).

Algorithm 1 consists of two stages; warm-up and main. The algorithm initiates the warm-up stage (Algorithm 2 in Appendix A.1) to apply regularization to the estimators, by uniform exploration

across all agents $n \in [N]$ for each arm $k \in [K]$. Subsequently, it proceeds to the main stage, which comprises multiple epochs denoted by $\tau$. In what follows, we describe the process for constructing assortments at each time step in the main stage. We first describe the procedure of estimation, assortments construction, and elimination associated with Lines 5-9 of Algorithm 1. For each $k \in [K]$, from observed feedback $y_{n,t} \in \{0,1\}$ for $n \in S_{k,t}$, $t \in \mathcal{T}_{k,\tau-1}$, where $\mathcal{T}_{k,\tau-1}$ is a set of the exploration time steps regarding arm $k$ before epoch $\tau$, we first define the negative log-likelihood loss as

$$l_{k,\tau-1}(\theta) = - \sum_{t \in \mathcal{T}_{k,\tau-1}} \sum_{n \in S_{k,t} \cup \{0\}} y_{n,t} \log p(n|S_{k,t}, \theta), \tag{1}$$

where, with a slight abuse of notation, $p(n|S_{k,t}, \theta) = \exp(z_n^\top \theta)/(1 + \sum_{m \in S_{k,t}} \exp(z_m^\top \theta))$. Then at the beginning of each epoch $\tau$, the algorithm estimates $\hat{\theta}_{k,\tau-1}$ from the method of Maximum Likelihood Estimation (MLE).

From the estimator, we define the upper and lower confidence bounds for the matching probability of assortment $S_k$ as

$$R_{k,\tau-1}^{UCB}(S_k) = \sum_{n \in S_k} \frac{\exp(p_{n,k,\tau-1})}{1 + \sum_{m \in S_k} \exp(p_{m,k,\tau-1})} \text{ and}$$

$$R_{k,\tau-1}^{LCB}(S_k) = \sum_{n \in S_k} \frac{\exp(b_{n,k,\tau-1})}{1 + \sum_{m \in S_k} \exp(b_{m,k,\tau-1})}, \tag{2}$$

where $p_{n,k,\tau-1} = z_n^\top \hat{\theta}_{k,\tau-1} + \beta_T \|z_n\|_{V_{k,\tau-1}^{-1}}$ and $b_{n,k,\tau-1} = z_n^\top \hat{\theta}_{k,\tau-1} - \beta_T \|z_n\|_{V_{k,\tau-1}^{-1}}$ with a confidence term $\beta_T = \frac{C_1}{\kappa}\sqrt{\log(TNK)}$ for some constant $C_1 > 0$ and $V_{k,\tau-1} = \sum_{t \in \mathcal{T}_{k,\tau-1}} \sum_{n \in S_{k,t}} z_n z_n^\top$.

Then the algorithm assesses the eligibility of each assignment $(n,k)$ for $n \in \mathcal{N}_{k,\tau-1}$ and $k \in [K]$ as a potential optimal assortment, where $\mathcal{N}_{k,\tau-1}$ is the active set of agents regarding arm $k$ at epoch $\tau$. To evaluate the assignment $(n,k)$, it constructs a representative assortment of $\{S_{l,\tau}^{(n,k)}\}_{l \in [K]}$ from an optimistic view as follows:

$$\{S_{l,\tau}^{(n,k)}\}_{l \in [K]} = \underset{\{S_k\}_{k \in [K]} \in \mathcal{M}_{\tau-1}: n \in S_k}{\arg\max} \sum_{k \in [K]} R_{k,\tau-1}^{UCB}(S_k). \tag{3}$$

Then based on the representative assortments, it obtains $\mathcal{N}_{k,\tau}$ by eliminating $n \in \mathcal{N}_{k,\tau-1}$ which satisfies the elimination condition in Line 9 of Algorithm 1 :

$$\max_{\{S_l\}_{l \in [K]} \in \mathcal{M}_{\tau-1}} \sum_{l \in [K]} R_{l,\tau-1}^{LBC}(S_l) > \sum_{l \in [K]} R_{l,\tau-1}^{UCB}(S_{l,\tau}^{(n,k)}), \tag{4}$$

in which $(n,k)$ is excluded if the reward from the optimistic assortment for $(n,k)$ is lower than that from the pessimistic optimal assortment. From the obtained $\mathcal{N}_{k,\tau}$ for all $k \in [K]$, it constructs an active set of assortments $\mathcal{M}_\tau$ (Line 17), which is likely to contain the optimal assortments as $\{S_k^*\}_{k \in [K]} \in \mathcal{M}_\tau$ (to be shown later).

Following the elimination process outlined above, here we describe the policy of assigning assortment $\{S_{k,t}\}_{k \in [K]}$ at each time $t$ corresponding to Lines 10-16 in Algorithm 1. For each $k \in [K]$, the algorithm utilizes the G/D-optimal design problem (Lattimore & Szepesvári, 2020) to obtain proportion $\pi_{k,\tau} \in \mathcal{P}(\mathcal{N}_{k,\tau})$ for learning $\theta_k^*$ efficiently by exploring agents in $\mathcal{N}_{k,\tau}$, where $\mathcal{P}(\mathcal{N}_{k,\tau})$ is the probability simplex with vertex set $\mathcal{N}_{k,\tau}$. Then, for all $n \in \mathcal{N}_{k,\tau}$, it explores $\{S_{l,\tau}^{(n,k)}\}_{l \in [K]}$ several times using $\pi_{k,\tau}(n)$ which is the corresponding value of $n$ in $\pi_{k,\tau}$. The algorithm repeats those processes over epochs $\tau$ until it reaches the time horizon $T$. In the following, we provide a regret bound of this algorithm.

**Theorem 1.** *Algorithm 1 achieves a regret bound of*

$$\mathcal{R}(T) = \tilde{\mathcal{O}}\left(\frac{1}{\kappa}K\sqrt{rKT}\right).$$

*Proof sketch.* Here we provide a proof sketch and the full version is provided in Appendix A.2. In this proof, we focus on the regret stemming from the main stage, as the regret incurred during the warm-up stage exhibits a poly-logarithmic bound regarding $T$, which is negligible for large $T$.

We first define a concentration bound event $E$ such that for all $\tau \in [T]$, $k \in [K]$, and $n \in [N]$, $|z_n^\top(\hat{\theta}_{k,\tau-1} - \theta_k^*)| \leq \frac{C_1}{\kappa}\sqrt{\|z_n\|_{V_{k,\tau-1}^{-1}}^2 \log(NKT)}$, for some constant $C_1 > 0$. Then we can show that $E$ holds with high probability. Therefore, for the following, we assume that $E$ holds. Then we show that for all $\tau \in [T]$,

$$\{S_k^*\}_{k\in[K]} \in \mathcal{M}_{\tau-1}. \tag{5}$$

This can be shown by induction. Suppose $\{S_k^*\}_{k\in[K]} \in \mathcal{M}_{\tau-1}$. Then, after omitting certain details, we can show that for any $n \in S_k^*$ and $k \in [K]$,

$$\sum_{l\in[K]} R_{l,\tau-1}^{UCB}(S_{l,\tau}^{(n,k)}) \geq \max_{\{S_l\}_{l\in[K]}\in\mathcal{M}_{\tau-1}} \sum_{l\in[K]} R_{l,\tau-1}^{LCB}(S_l), \tag{6}$$

which implies that $n \in S_k^*$ is not eliminated from the elimination condition in Line 9 of Algorithm 1 so that $\{S_k^*\}_{k\in[K]} \in \mathcal{M}_\tau$. With $\{S_k^*\}_{k\in[K]} \in \mathcal{M}_0$, this concludes the induction.

By using the mean value theorem, we can show that for any $S \subset \mathcal{N}_{k,\tau-1}$ for $k \in [K]$ and $\tau \in [T]$ we have

$$R_{k,\tau-1}^{UCB}(S) - R_k(S) \leq 2\beta_T \max_{n\in S}\|z_n\|_{V_{k,\tau-1}^{-1}} \text{ and } R_k(S) - R_{k,\tau-1}^{LCB}(S) \leq 2\beta_T \max_{n\in S}\|z_n\|_{V_{k,\tau-1}^{-1}}. \tag{7}$$

Then we have

$$\sum_{l\in[K]} R_l(S_l^*) - \sum_{l\in[K]} R_l(S_{l,\tau}^{(n,k)})$$

$$\leq \sum_{l\in[K]} \left( R_{l,\tau-1}^{LCB}(S_l^*) + 2\beta_T \max_{m\in S_l^*}\|z_m\|_{V_{l,\tau-1}^{-1}} - R_{l,\tau-1}^{UCB}(S_{l,\tau}^{(n,k)}) + 2\beta_T \max_{m\in S_{l,\tau}^{(n,k)}}\|z_m\|_{V_{l,\tau-1}^{-1}} \right)$$

$$\leq 2\beta_T \sum_{l\in[K]} (\max_{m\in S_l^*}\|z_m\|_{V_{l,\tau-1}^{-1}} + \max_{m\in S_{l,\tau}^{(n,k)}}\|z_m\|_{V_{l,\tau-1}^{-1}}), \tag{8}$$

where the first inequality comes from (7) and the last one comes from $\sum_{l\in[K]} R_{l,\tau-1}^{LCB}(S_l^*) \leq \sum_{l\in[K]} R_{l,\tau-1}^{UCB}(S_{l,\tau}^{(n,k)})$, which can be derived from (5) and (6).

Now we provide a bound for $\|z_m\|_{V_{k,\tau-1}^{-1}}$ in the last inequality of (8). Define $V(\pi_{k,\tau}) = \sum_{n\in\mathcal{N}_{k,\tau}} \pi_{k,\tau}(n)z_n z_n^\top$. From the algorithm, we can observe that

$$V_{k,\tau} \succeq \sum_{n\in\mathcal{N}_{k,\tau}} r\pi_{k,\tau}(n)T_\tau z_n z_n^\top \succeq C_2 2^{\tau-1} r \log(KNT)V(\pi_{k,\tau}), \tag{9}$$

where the first and second inequalities are obtained from the definition of $V_{k,\tau}$ and $T_\tau$, respectively. From Theorem 21.1 (Kiefer-Wolfowitz) in Lattimore & Szepesvári (2020) for the G/D-optimal design problem, for all $k \in [K]$, we have $\max_{n\in\mathcal{N}_{k,\tau}}\|z_n\|_{V(\pi_{k,\tau})^{-1}}^2 = r$. Then, for any $n \in \mathcal{N}_{k,\tau}$ we have

$$\|z_n\|_{V_{k,\tau}^{-1}} = \mathcal{O}\left(2^{-\tau/2}\sqrt{\|z_n\|_{V(\pi_{\tau,l})^{-1}}^2/r\log(KNT)}\right) = \mathcal{O}(2^{-\tau/2}/\sqrt{\log(KNT)}), \tag{10}$$

where the first equality comes from (9). Therefore from (8) and (10), we have

$$\sum_{l\in[K]} (R_l(S_l^*) - R_l(S_{l,\tau}^{(n,k)})) = \mathcal{O}\left(\frac{1}{\kappa}K2^{-\tau/2}\right). \tag{11}$$

We define $\tau^*$ to be the smallest $\tau \in [T]$ such that $\sum_{k \in [K]} |\mathcal{T}_{k,\tau}^{(2)}| \geq K \sum_{\tau'=1}^{\tau} 2^{\tau'-1} C_2 r \log(KT) \geq T$. Then we have $\sum_{\tau'=1}^{\tau^*} 2^{\tau'} = \Theta(T/rK\log(KNT))$, which implies $\tau^* = \mathcal{O}(\log(T/rK\log(KNT)))$. Finally, we can conclude the proof from the following:

$$\mathcal{R}(T) = \mathbb{E}\left[\sum_{t \in [T]} \sum_{k \in [K]} R_k(S_k^*) - R_k(S_{k,t})\right]$$

$$= \widetilde{\mathcal{O}}\left(\mathbb{E}\left[\sum_{\tau=1}^{\tau^*} \sum_{k \in [K]} \sum_{n \in \mathcal{N}_{k,\tau}} |\mathcal{T}_{n,k,\tau}^{(2)}| \times \sum_{l \in [K]} R_l(S_l^*) - R_l(S_{l,\tau}^{(n,k)})\right]\right)$$

$$= \widetilde{\mathcal{O}}\left(\mathbb{E}\left[\frac{1}{\kappa} K \sum_{\tau=1}^{\tau^*} \sum_{k \in [K]} \sum_{n \in \mathcal{N}_{k,\tau}} |\mathcal{T}_{n,k,\tau}^{(2)}| 2^{-\tau/2}\right]\right)$$

$$= \widetilde{\mathcal{O}}\left(\mathbb{E}\left[\frac{1}{\kappa} K^2 r \sum_{\tau=1}^{\tau^*} 2^{\tau/2}\right]\right) = \widetilde{\mathcal{O}}\left(\frac{1}{\kappa} K\sqrt{rKT}\right),$$

where the third equality comes from (11) and the last one comes from the bound of $\tau^*$. $\qquad\square$

Algorithm 1 shows sublinear regret with respect to $r (\leq d)$ instead of $d$ from the procedure of SVD. Moreover, the algorithm demonstrates a tight regret bound concerning $T$ and $r$. We also observe that same as the previous linear-utility MNL bandits in Oh & Iyengar (2021; 2019), our regret bound depends on $1/\kappa$ which is bounded by $O(L^2)$ in the worst case. It remains an open question whether the dependence on $\kappa$ can be further improved.

We can observe that there may exist a computation issue in the combinatorial optimization of our algorithm, specifically in (3) and (4), even though the computation is required for each epoch rather than each round. We address this issue in the following.

## 5 COMBINATORIAL OPTIMIZATION WITH $\alpha$-APPROXIMATION ORACLE

Here we discuss the combinatorial optimization in our algorithm. The exact optimization regarding (3) and (4) can of course be expensive due to its NP-hard nature. To address this, we can utilize an $\alpha$-approximation oracle with $0 \leq \alpha \leq 1$, first introduced in Kakade et al. (2007). Instead of obtaining the exact optimal assortment solution, the $\alpha$-approximation oracle, denoted by $\mathbb{O}^\alpha$, outputs $\{S_k^\alpha\}_{k \in [K]}$ satisfying $\sum_{k \in [K]} f_k(S_k^\alpha) \geq \max_{\{S_k\}_{k \in [K]} \in \mathcal{M}} \sum_{k \in [K]} \alpha f_k(S_k)$. Such an oracle can be constructed using a straightforward greedy policy as outlined in prior work (Kapralov et al., 2013; Calinescu et al., 2011).

We introduce an algorithm (Algorithm 3 in Appendix A.3) by modifying Algorithm 1 to incorporate $\alpha$-approximation oracles for the optimization. Due to the page limitation, here we explain only the distinct parts of the algorithm. For testing the assignment $(n, k)$, the algorithm constructs assortment $\{S_{l,\tau}^{\alpha,(n,k)}\}_{l \in [K]}$ (where $n \in S_{k,\tau}^{\alpha,(n,k)}$) in an optimistic view with an $\alpha$-approximation oracle to resolve computation issue as follows. We define an approximation oracle $\mathbb{O}_{UCB}^{\alpha,(n,k)}$ which outputs $\{S_{l,\tau}^{\alpha,(n,k)}\}_{l \in [K]}$ satisfying

$$\max_{\{S_l\}_{l \in [K]} \in \mathcal{M}_{\tau-1}:n \in S_k} \sum_{l \in [K]} \alpha R_{l,\tau-1}^{UCB}(S_l) \leq \sum_{l \in [K]} R_{l,\tau-1}^{UCB}(S_{l,\tau}^{\alpha,(n,k)}), \tag{12}$$

which replaces Line 8 in Algorithm 1. For the elimination procedure, we define another $\alpha$-approximation oracle, denoted by $\mathbb{O}_{LCB}^\alpha$, which outputs $\{S_{l,\tau}^\alpha\}_{l \in [K]}$ satisfying

$$\max_{\{S_l\}_{l \in [K]} \in \mathcal{M}_{\tau-1}} \sum_{l \in [K]} \alpha R_{l,\tau-1}^{LCB}(S_l) \leq \sum_{l \in [K]} R_{l,\tau-1}^{LCB}(S_{l,\tau}^\alpha). \tag{13}$$

Then it updates $\mathcal{N}_{k,\tau}$ by eliminating $n \in \mathcal{N}_{k,\tau-1}$ which satisfies the elimination condition of

$$\sum_{l \in [K]} \alpha R_{l,\tau-1}^{LCB}(S_{l,\tau}^\alpha) > \sum_{l \in [K]} R_{l,\tau-1}^{UCB}(S_{l,\tau}^{\alpha,(n,k)}), \tag{14}$$

which replaces Line 9 in Algorithm 1. We note that the algorithm utilizes the two different types of approximation oracles, $\mathbb{O}_{UCB}^{\alpha,(n,k)}$ and $\mathbb{O}_{LCB}^{\alpha}$. Then the algorithm achieves a regret bound for $\gamma$-regret defined as $\mathcal{R}^{\gamma}(T) = \mathbb{E}[\sum_{t\in[T]}\sum_{k\in[K]}\gamma R_k(S_k^*) - R_k(S_{k,t})]$ in the following theorem.

**Theorem 2.** *Algorithm 3 achieves a regret bound with $\gamma = \alpha^2$ as*

$$\mathcal{R}^{\gamma}(T) = \widetilde{\mathcal{O}}\left(\frac{1}{\kappa}K\sqrt{rKT}\right).$$

*Proof.* The proof is provided in Appendix A.3. □

## 6 EXPERIMENTS

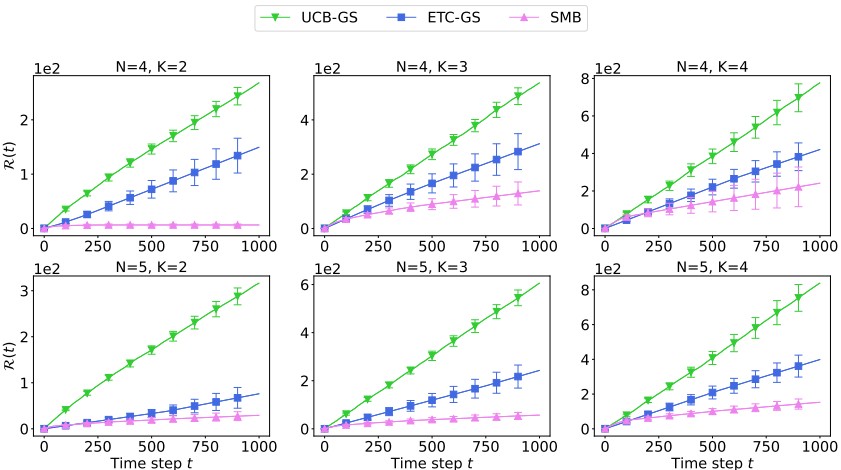

Figure 2: Experimental results for regret of algorithms

In this section, we present the results of experiments conducted with synthetic datasets to showcase the performance of our proposed algorithm. For these synthetic datasets, we randomly generate $\theta_k \in \mathbb{R}^d$ for $k \in [K]$ and $x \in \mathbb{R}^d$ drawing each element from a uniform distribution $[-1, 1]$ and subsequently normalizing them. The experimental setup involves fixed parameters such as $L = 2$, and $d = 3$ with variations in $N$ and $K$. We note that $r = d$ with probability 1 because each element in features is generated from the uniform distribution and $N > d$.

Unfortunately, no dedicated benchmark exists for our stochastic matching bandit scenario. Therefore, we choose to compare our algorithm (SMB) against Explore-then-Commit with Gale-Shapley (ETC-GS) and UCB with Gale-Shapley (UCB-GS), proposed for matching bandits by Liu et al. (2020). In our adoption of these algorithms, it's important to note that they require information about the preferences of arms over agents, which remains unknown in our setting. Therefore, we employ estimated values for these preferences at each time step. In Figure 2, it is evident that our algorithm surpasses the benchmarks for various settings. We also include an experimental result using features from a Gaussian distribution in Appendix A.5.

## 7 CONCLUSION

In this work, we consider a novel framework of stochastic matching bandits employing the MNL choice model with features. We propose an elimination-based algorithm that achieves a regret of $\widetilde{\mathcal{O}}(K\sqrt{rKT})$. We also discuss adopting $\alpha$-approximation oracle in our algorithm to handle computation issues related to combinatorial optimization. Lastly, we demonstrate the performance of our algorithms through experiments on synthetic datasets.

**Reproducibility Statement.** Source code is submitted as supplementary material and complete proofs of the theorems are included in the appendix.

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

## A    APPENDIX

### A.1    WARM-UP STAGE FOR ALGORITHM 1

Let $\lambda_{\min}(A)$ denote the minimum eigenvalue of matrix $A$. Then we provide the warm-up stage for Algorithm 1 in Algorithm 2.

---

**Algorithm 2** `Round-robin Warm-up`

---

$\lambda_{\min} \leftarrow \lambda_{\min}(\sum_{n \in [N]} z_n z_n^\top)$
**for** $k \in [K]$ **do**
    $t_k \leftarrow t, i \leftarrow \min\{L, N\}$
    $T_k \leftarrow (C_3 N/i\kappa^2 \lambda_{\min} \log(TK))(r + \log(TK))^2$
    $\mathcal{T}_k^{(1)} \leftarrow [t_k, t_k + T_k - 1]$
    **for** $t \in \mathcal{T}_k^{(1)}$ **do**
        $a \leftarrow (i(t-1) + 1 \mod N), b \leftarrow (it \mod N)$
        **if** $a \leq b$ **then**
            $S_{k,t} \leftarrow [a, b]$
        **else**
            $S_{k,t} \leftarrow [1, b] \cup [a, N]$
        Construct any $S_{l,t}$ for $l \in [K]/\{k\}$ satisfying $\{S_{k,t}\}_{k \in [K]} \in \mathcal{M}_0$
        Offer $\{S_{k,t}\}_{k \in [K]}$ and observe feedback $y_{n,t} \in \{0, 1\}$ for all $n \in S_{k,t}, k \in [K]$

---

### A.2    PROOF OF THEOREM 1

In the following proof, with a slight abuse of notation, we use $p(n|S, \theta) = \exp(z_n^\top \theta)/(1 + \sum_{m \in S} \exp(z_m^\top \theta))$. We provide a lemma for a confidence bound.

**Lemma 1.** *For any $\tau \in [T]$, $k \in [K]$, and $n \in [N]$, with probability at least $1 - \delta$, for some constant $C > 0$, we have*

$$|z_n^\top (\hat{\theta}_{k,\tau-1} - \theta_k^*)| \leq (C/\kappa)\sqrt{\|z_n\|_{V_{k,\tau-1}^{-1}}^2 \log(TKN/\delta)}.$$

*Proof.* We first provide a bound in the following lemma.

**Lemma 2.** *For any $n \in [N]$, $k \in [K]$, and $\tau \in [T]$, with probability at least $1 - \delta$, we have*

$$|z_n^\top (\hat{\theta}_{k,\tau-1} - \theta_k^*)| \leq \frac{2\sqrt{\log(TKN/\delta)}}{\kappa} \|z_n\|_{V_{k,\tau}^{-1}} + \frac{6}{\kappa^2} \|\hat{\theta}_{k,\tau} - \theta_k^*\|_2 \|g_{k,\tau}(\hat{\theta}_{k,\tau}) - g_{k,\tau}(\theta_k^*)\|_{V_{k,\tau}^{-1}} \|z_n\|_{V_{k,\tau}^{-1}}.$$

*Proof.* The proof is deferred to Appendix A.4.1           $\square$

Then we define

$$E_1 = \left\{ |z_n^\top (\hat{\theta}_{k,\tau-1} - \theta_k^*)| \leq \frac{2\sqrt{\log(TKN/\delta)}}{\kappa} \|z_n\|_{V_{k,\tau}^{-1}} \right.$$
$$\left. + \frac{6}{\kappa^2} \|\hat{\theta}_{k,\tau} - \theta_k^*\|_2 \|g_{k,\tau}(\hat{\theta}_{k,\tau}) - g_{k,\tau}(\theta_k^*)\|_{V_{k,\tau}^{-1}} \|z_n\|_{V_{k,\tau}^{-1}} \, \forall n \in [N], k \in [K], \tau \in [T] \right\},$$

which holds at least $1 - \delta$. Now we provide bounds for $\|\hat{\theta}_{k,\tau} - \theta_k^*\|_2$ and $\|g_{k,\tau}(\hat{\theta}_{k,\tau}) - g_{k,\tau}(\theta_k^*)\|_{V_{k,\tau}^{-1}}$.

**Lemma 3** (Lemma 7 in Li et al. (2017))**.** *For all $k \in [K]$, $\tau \in [T]$, with probability at least $1 - \delta$ for $\delta > 0$, we have*

$$\|g_{k,\tau-1}(\hat{\theta}_{k,\tau-1}) - g_{k,\tau-1}(\theta_k^*)\|_{V_{k,\tau-1}^{-1}} \leq 4\sqrt{2r + \log(KTN/\delta)}.$$

We define $V_k^0 = \sum_{t \in \mathcal{T}_k^{(1)}} \sum_{n \in S_{k,t}} z_n z_n^\top$. Then we have the following lemma.

**Lemma 4.** *For all $k \in [K]$, we have $\lambda_{\min}(V_k^0) \geq (C_0/\kappa^2 \log(TKN/\delta))(r^2 + \log^2(TKN/\delta) + 2r \log(TKN/\delta))$.*

*Proof.* Let $\lambda' = (C_0/\kappa^2 \lambda_{\min} \log(TK/\delta))(r^2 + \log^2(TKN/\delta) + 2r \log(TKN/\delta))$ and recall $\lambda_{\min} = \lambda_{\min}(\sum_{n \in [N]} z_n z_n^\top)$. From the phase in the warm-up stage (Algorithm 2), we can observe that $V_k^0$ contains $z_n z_n^\top$ for each $n \in [N]$ at least $\lambda'$. Since $\sum_{n \in [N]} z_n z_n^\top = \sum_{s \in [r]} \lambda_s u_s u_s^\top$, we have $V_k^0 = \sum_{t \in \mathcal{T}_k^{(1)}} \sum_{n \in S_{k,t}} z_n z_n^\top = \sum_{s \in [r]} \lambda_s' u_s u_s^\top$ where $\lambda_s' \geq \lambda' \lambda_s$. Then from $\lambda_{\min} = \lambda_r$, we can conclude $\lambda_{\min}(V_k^0) \geq \lambda' \lambda_{\min}$. $\square$

**Lemma 5** (Lemma 9 in Kveton et al. (2020)). *Suppose $\lambda_{\min}(V_k^0) \geq \max\{(1/4\kappa^2)(r \log(T/r) + 2\log(KTN/\delta)), 1\}$ for all $k \in [K]$. Then, for all $\tau \in [T]$ and $k \in [K]$, we have*

$$\mathbb{P}(\|\hat{\theta}_{k,\tau-1} - \theta_k^*\|_2 \geq 1) \leq 1/\delta.$$

We define $E_2 = \{\|\hat{\theta}_{k,\tau-1} - \theta_k^*\|_2 \leq 1 \; \forall k \in [K], \tau \in [T]\}$. Then from Lemmas 4, 5, we have $\mathbb{P}(E_1) \geq 1 - \delta$.

We also denote by $E_3$ the event of $\{\|g_{k,\tau-1}(\hat{\theta}_{k,\tau-1}) - g_{k,\tau-1}(\theta_k^*)\|_{V_{k,\tau-1}^{-1}} \leq 4\sqrt{2r + \log(KTN/\delta)} \; \forall \tau \in [T], k \in [K]\}$, which hold with probability at least $1 - \delta$ from Lemma 3.

**Lemma 6.** *Under $E_2$ and $E_3$, for any $\tau \in [T]$, $k \in [K]$, we have*

$$\|\hat{\theta}_{k,\tau-1} - \theta_k^*\|_2 \leq \frac{2}{\kappa} \sqrt{\frac{2r + \log(TNK/\delta)}{\lambda_{\min}(V_k^0)}}.$$

*Proof.* The proof is deferred to Appendix A.4.2

$\square$

Finally, under $E_1 \cup E_2 \cup E_3$ which holds with probability at least $1 - 3\delta$, we have

$$|z_n^\top(\hat{\theta}_{k,\tau} - \theta_k^*)|$$
$$\leq \frac{2\sqrt{\log(TKN/\delta)}}{\kappa} \|z_n\|_{V_{k,\tau}^{-1}} + (6/\kappa^2)\|z_n\|_{V_{k,\tau}^{-1}} \|\hat{\theta}_k^* - \theta_k^*\|_2 \|(g_{k,\tau}(\hat{\theta}_{k,\tau}) - g_{k,\tau}(\theta_k^*))\|_{V_{k,\tau}^{-1}}$$
$$\leq \frac{2\sqrt{\log(TKN/\delta)}}{\kappa} \|z_n\|_{V_{k,\tau}^{-1}} + \frac{48(2r + \log(KTN/\delta))}{\kappa^2 \sqrt{\lambda_{\min}(V_{k,\tau}^0)}} \|z_n\|_{V_{k,\tau}^{-1}}$$
$$\leq \frac{3\sqrt{\log(TKN/\delta)}}{\kappa} \|z_n\|_{V_{k,\tau}^{-1}}$$
$$= (3/\kappa)\sqrt{\|z_n\|_{V_{\tau,k}^{-1}}^2 \log(TKN/\delta)},$$

which concludes the proof. $\square$

Then we define event $E = \{|z_n^\top(\hat{\theta}_{k,\tau} - \theta_k^*)| \leq (C_1/\kappa)\sqrt{\|z_n\|_{V_{k,\tau}^{-1}}^2 \log(TKN)} \; \forall \tau \in [T], k \in [K], n \in [N]\}$ for some $C_1 > 0$, which holds at least $1 - 1/T$ with Lemma 1 and $\delta = 1/T$. In the following, we provide a lemma for showing that $\mathcal{M}_\tau$ is likely to contain the optimal assortment.

**Lemma 7.** *Under $E$, $(S_1^*, \ldots, S_K^*) \in \mathcal{M}_{\tau-1}$ for all $\tau \in [T]$.*

*Proof.* Here we use induction for the proof. Suppose that for fixed $\tau$, we have $(S_1^*, \ldots, S_K^*) \in \mathcal{M}_\tau$ for all $k \in [K]$. Recall that $\beta_T = (C_1/\kappa)\sqrt{\log(TKN)}$. Since $x/(1+x)$ is a non-decreasing function for $x > -1$ and $z_n^\top \hat{\theta}_{k,\tau} - \beta_T \|z_n\|_{V_{k,\tau}^{-1}} \leq z_n^\top \theta_k^* \leq z_n^\top \hat{\theta}_{k,\tau} + \beta_T \|z_n\|_{V_{k,\tau}^{-1}}$ from $E$, we have

$R_{k,\tau}^{UCB}(S) \geq R_k(S)$ and $R_{k,\tau}^{LCB}(S) \leq R_k(S)$ for any $S \subset [N]$. Then for $k \in [K]$, $n \in S_k^*$, and any $(S_1, .., S_K) \in \mathcal{M}_\tau$, we have

$$\sum_{l \in [K]} R_{l,\tau}^{UCB}(S_{l,\tau+1}^{(n,k)}) \geq \sum_{l \in [K]} R_{l,\tau}^{UCB}(S_l^*)$$

$$\geq \sum_{l \in [K]} R_l(S_l^*)$$

$$\geq \sum_{l \in [K]} R_l(S_l)$$

$$\geq \sum_{l \in [K]} R_{l,\tau}^{LCB}(S_l), \tag{15}$$

where the first inequality comes from the elimination condition in the algorithm and $(S_1^*, \ldots S_K^*) \in \mathcal{M}_\tau$, and the third inequality comes from the optimality of $(S_1^*, \ldots, S_K^*)$. This implies that $n \in \mathcal{N}_{k,\tau+1}$ from the algorithm. Then by following the same statement of (15) for all $n \in S_k^*$ and $k \in [K]$, we have $S_k^* \subset \mathcal{N}_{k,\tau+1}$ for all $k \in [K]$, which implies $(S_1^*, \ldots, S_K^*) \in \mathcal{M}_{\tau+1}$. Therefore, with $(S_1^*, \ldots, S_K^*) \in \mathcal{M}_1$, we can conclude the proof from the induction. $\square$

**Lemma 8.** *Under $E$, for any $S \subset \mathcal{N}_{k,\tau-1}$ for all $k \in [K]$ and $\tau \in [T]$ we have*

$$R_{k,\tau-1}^{UCB}(S) - R_k(S) \leq 2\beta_T \max_{n \in S} \|z_n\|_{V_{k,\tau-1}^{-1}} \text{ and}$$

$$R_k(S) - R_{k,\tau-1}^{LCB}(S) \leq 2\beta_T \max_{n \in S} \|z_n\|_{V_{k,\tau-1}^{-1}}.$$

*Proof.* For the proof, we follow the proof steps in Lemma 5 in Oh & Iyengar (2021). Let $u_{n,k} = z_n^\top \theta_k^*$. From $E$, we can observe that $b_{n,k,\tau-1} \leq u_{n,k} \leq p_{n,k,\tau-1}, p_{n,k,\tau-1} - u_{n,k} \leq 2\beta_T \|z_n\|_{V_{k,\tau}^{-1}}$, and $u_{n,k} - b_{n,k,\tau-1} \leq 2\beta_T \|z_n\|_{V_{k,\tau}^{-1}}$. Then by the mean value theorem, there exists $\bar{u}_{n,k} = (1 - c)p_{n,k,\tau-1} + cu_{n,k}$ for some $c \in (0,1)$ satisfying, for any $S \subset \mathcal{N}_{k,\tau-1}$

$$R_{k,\tau-1}^{UCB}(S) - R_k(S) = \frac{\sum_{n \in S} \exp(p_{n,k,\tau-1})}{1 + \sum_{m \in S} \exp(p_{m,k,\tau-1})} - \frac{\sum_{n \in S} \exp(u_{n,k})}{1 + \sum_{m \in S} \exp(u_{m,k})}$$

$$= \sum_{n \in S} \nabla_{v_n} \left( \frac{\sum_{m \in S} \exp(v_m)}{1 + \sum_{m \in S} \exp(v_m)} \right) \Big|_{v_n = \bar{u}_{n,k}} (p_{n,k,\tau-1} - u_{n,k})$$

$$= \frac{(1 + \sum_{n \in S} \exp(\bar{u}_{n,k}))(\sum_{n \in S} \exp(\bar{u}_{n,k})(p_{n,k,\tau-1} - u_{n,k}))}{(1 + \sum_{n \in S} \exp(\bar{u}_{n,k}))^2}$$

$$- \frac{(\sum_{n \in S} \exp(\bar{u}_{n,k}))(\sum_{n \in S} \exp(\bar{u}_{n,k})(p_{n,k,\tau-1} - u_{n,k}))}{(1 + \sum_{n \in S} \exp(\bar{u}_{n,k}))^2}$$

$$\leq \sum_{n \in S} \frac{\exp(\bar{u}_{n,k})}{1 + \sum_{m \in S} \exp(\bar{u}_{m,k})} (p_{n,k,\tau-1} - u_{n,k})$$

$$\leq \max_{n \in S}(p_{n,k,\tau-1} - u_{n,k})$$

$$\leq 2(C_1/\kappa)\sqrt{\log(KNT)} \max_{n \in S} \|z_n\|_{V_{k,\tau-1}^{-1}}.$$

Following the same procedure, there exists $\bar{u}_{n,k} = (1 - c)u_{n,k} + cb_{n,k,\tau-1}$ for some $c \in (0,1)$ satisfying

$$R_k(S) - R_{k,\tau-1}^{LCB}(S_k) \leq \sum_{n \in S} \frac{\sum_{n \in S} \exp(\bar{u}_{n,k})}{1 + \sum_{n \in S} \exp(\bar{u}_{n,k})} (u_{n,k} - b_{n,k,\tau-1})$$

$$\leq \max_{n \in S}(u_{n,k} - b_{n,k,\tau-1})$$

$$\leq 2(C_1/\kappa)\sqrt{\log(KNT)} \max_{n \in S} \|z_n\|_{V_{k,\tau-1}^{-1}},$$

which concludes the proof. $\square$

From the above Lemmas 7 and 8, under $E$, we have

$$\sum_{l\in[K]} R_l(S_l^*) - \sum_{l\in[K]} R_l(S_{l,\tau}^{(n,k)}) \le \sum_{l\in[K]} R_{l,\tau-1}^{LCB}(S_l^*) + 2\beta_T \max_{m\in S_l^*} \|z_m\|_{V_{l,\tau-1}^{-1}}$$

$$- \sum_{l\in[K]} R_{l,\tau-1}^{UCB}(S_{l,\tau}^{(n,k)}) + 2\beta_T \max_{m\in S_{l,\tau}^{(n,k)}} \|z_m\|_{V_{l,\tau-1}^{-1}}$$

$$\le 2\beta_T \sum_{l\in[K]} (\max_{m\in S_l^*} \|z_m\|_{V_{l,\tau-1}^{-1}} + \max_{m\in S_{l,\tau}^{(n,k)}} \|z_m\|_{V_{l,\tau-1}^{-1}}), \quad (16)$$

where the last inequality comes from the fact that $(S_1^*, \ldots, S_K^*) \in \mathcal{M}_{\tau-1}$ and $\max_{(S_1,\ldots,S_K)\in\mathcal{M}_{\tau-1}} \sum_{l\in[K]} R_{l,\tau-1}^{LCB}(S_l) \le \sum_{l\in[K]} R_{l,\tau-1}^{UCB}(S_{l,\tau}^{(n,k)})$ from the algorithm.

We define $V(\pi_{k,\tau}) = \sum_{n\in\mathcal{N}_{k,\tau}} \pi_{k,\tau}(n) z_n z_n^\top$ and $supp(\pi_{k,\tau}) = \{n \in \mathcal{N}_{k,\tau} : \pi_{k,\tau}(n) \ne 0\}$. Then we have the following lemma from the G/D-optimal design problem.

**Lemma 9** (Theorem 21.1 (Kiefer-Wolfowitz) in Lattimore & Szepesvári (2020)). *For all $\tau \in [T]$ and $k \in [K]$, we have*

$$\max_{n\in\mathcal{N}_{k,\tau}} \|z_n\|_{V(\pi_{k,\tau})^{-1}}^2 = r \text{ and } |supp(\pi_{k,\tau})| \le r(r+1)/2.$$

From the definition of $V_{k,\tau}$ and $T_\tau$, we have

$$V_{k,\tau} \succeq \sum_{\tau'=1}^{\tau} \sum_{n\in\mathcal{N}_{k,\tau'}} r\pi_{k,\tau'}(n) T_{\tau'} z_n z_n^\top$$

$$\succeq C_2 2^{\tau-1} r \log(KNT) V(\pi_{k,\tau}). \quad (17)$$

Then from Lemma 9 and (17), for any $n \in \mathcal{N}_{k,\tau}$ we have

$$\beta_T \|z_n\|_{V_{k,\tau}^{-1}} = (1/\kappa)\sqrt{\|z_n\|_{V_{k,\tau}^{-1}}^2 \log(KNT)}$$

$$= \mathcal{O}\left((1/\kappa)2^{-\tau/2}\sqrt{\|z_n\|_{V(\pi_{\tau,l})^{-1}}^2/r}\right)$$

$$= \mathcal{O}((1/\kappa)2^{-\tau/2}). \quad (18)$$

Therefore under $E$, from (16) and (18), for $\tau > 1$, we have

$$\sum_{l\in[K]} (R_l(S_l^*) - R_l(S_{l,\tau}^{(n,k)})) = \mathcal{O}((1/\kappa)K 2^{-\tau/2}).$$

We define $\tau^*$ is the smallest $\tau \in [T]$ such that

$$\sum_{k\in[K]} |\mathcal{T}_{k,\tau}^{(2)}| \ge \sum_{\tau'=1}^{\tau} \sum_{k\in[K]} 2^{\tau'-1}(1/\kappa) C_2 r \log(KNT) \ge T.$$

Then we have $\sum_{\tau'=1}^{\tau^*} \sum_{k\in[K]} 2^{\tau'} = \Theta(T/r\log(KNT))$, which implies $\tau^* = \mathcal{O}(\log(T/rK\log(KNT)))$.

We have

$$\mathcal{R}(T) = \mathbb{E}\left[\sum_{t\in[T]} \sum_{k\in[K]} R_k(S_k^*) - R_k(S_{k,t})\right]$$

$$\le \mathbb{E}\left[\sum_{l\in[K]} \sum_{t\in\mathcal{T}_l^{(1)}} \sum_{k\in[K]} R_k(S_k^*) - R_k(S_{k,t}) + \sum_{l\in[K]} \sum_{t\in\mathcal{T}_{l,\tau^*}^{(2)}} \sum_{k\in[K]} R_k(S_k^*) - R_k(S_{k,t})\right],$$

$$(19)$$

which consists of regret from the stage of warming up and main. We first analyze the regret from the warming-up as follows:

$$
\mathbb{E}\left[\sum_{l\in[K]}\sum_{t\in\mathcal{T}_l^{(1)}}\sum_{k\in[K]} R_k(S_k^*) - R_k(S_{k,t})\right] \le \mathbb{E}\left[\sum_{l\in[K]} KT_l\right]
$$

$$
= \widetilde{\mathcal{O}}(r^2 K^2 N/(\min\{L,N\}\kappa^2\lambda_{\min})). \tag{20}
$$

For the regret bound from the main part of the algorithm, with large enough $T$, we have

$$
\mathbb{E}\left[\sum_{l\in[K]}\sum_{t\in\mathcal{T}_{l,\tau^*}^{(2)}}\sum_{k\in[K]} R_k(S_k^*) - R_t(S_{k,t})\right]
$$

$$
\le \mathbb{E}\left[\sum_{l\in[K]}\sum_{t\in\mathcal{T}_{l,\tau^*}^{(2)}}\sum_{k\in[K]} (R_k(S_k^*) - R_k(S_{k,t}))\,\mathbb{1}(E)\right]
$$

$$
+ \mathbb{E}\left[\sum_{l\in[K]}\sum_{t\in\mathcal{T}_{l,\tau^*}^{(2)}}\sum_{k\in[K]} (R_k(S_k^*) - R_k(S_{k,t}))\,\mathbb{1}(E^c)\right]
$$

$$
= \mathcal{O}\left((K/\kappa)\sum_{\tau=1}^{\tau^*}\sum_{l\in[K]}\sum_{n\in\mathcal{N}_{l,\tau}} |\mathcal{T}_{n,l,\tau}^{(2)}|2^{-\tau/2}\right) + \mathcal{O}(rK\log(NKT)) + \mathcal{O}(K)
$$

$$
= \mathcal{O}\left((K/\kappa)\sum_{\tau=1}^{\tau^*}\sum_{l\in[K]}\sum_{n\in\mathcal{N}_{l,\tau}} |\mathcal{T}_{n,l,\tau}^{(2)}|2^{-\tau/2}\right)
$$

$$
= \widetilde{\mathcal{O}}\left((K/\kappa)\sum_{\tau=1}^{\tau^*}\sum_{l\in[K]} (r2^\tau + |Supp(\pi_{l,\tau})|)2^{-\tau/2}\right)
$$

$$
= \widetilde{\mathcal{O}}\left((K^2/\kappa)\sum_{\tau=1}^{\tau^*}(r2^{\tau/2} + r^2 2^{-\tau/2})\right)
$$

$$
= \widetilde{\mathcal{O}}\left((K^2/\kappa)(r2^{\tau^*/2} + r^2)\right)
$$

$$
= \widetilde{\mathcal{O}}\left((K/\kappa)\sqrt{KrT}\right), \tag{21}
$$

where the third last equality comes from Lemma 9 and the last equality comes from $\tau^* = \widetilde{\Theta}(\log(T/rK))$. From (19), (20), (21), for large enough $T$, we can conclude that

$$
\mathcal{R}(T) = \widetilde{\mathcal{O}}\left((K/\kappa)\sqrt{KrT}\right).
$$

## A.3 $\alpha$-APPROXIMATION ORACLE

### A.3.1 ALGORITHM

---

**Algorithm 3** Elimination-based Stochastic Matching Bandit with $\alpha$-Approximation Oracle

---

**Input:** $N, L, K, T, \kappa, C_1 > 0, C_2 > 0, C_3 > 0$

**Init:** $t \leftarrow 1, T_1 \leftarrow C_2 \log(NKT), \mathcal{N}_{k,0} \leftarrow [N], \mathcal{T}_{n,k,0}^{(2)} = \varnothing$ for all $k \in [K], n \in [N]$

19  Find SVD of $X = U\Sigma V^\top$ and obtain $U_r = [u_1, \ldots, u_r]$

20  $z_n \leftarrow U_r^\top x_n$ for $n \in [N]$

21  Run $\texttt{Warm-up}$ (Algorithm 2) over time steps in $\mathcal{T}_k^{(1)}$ (defined in Algorithm 2) for $k \in [K]$

22  **for** $\tau = 1, 2 \ldots$ **do**

 // Estimation

23   $\mathcal{T}_{k,\tau-1} \leftarrow \mathcal{T}_k^{(1)} \cup \mathcal{T}_{k,\tau-1}^{(2)}$ for $k \in [K]$ where $\mathcal{T}_{k,\tau-1}^{(2)} := \bigcup_{\tau' \in [\tau-1]} \bigcup_{n \in \mathcal{N}_{k,\tau'}} \mathcal{T}_{n,k,\tau'}^{(2)}$

24   $V_{k,\tau-1} \leftarrow \sum_{s \in \mathcal{T}_{k,\tau-1}} \sum_{n \in S_{k,s}} z_n z_n^\top$ for $k \in [K]$

25   $\hat{\theta}_{k,\tau-1} \leftarrow \arg\min_{\theta \in \mathbb{R}^r} l_{k,\tau-1}(\theta)$ for $k \in [K]$ where $l_{k,\tau-1}(\theta) :=$ $-\sum_{s \in \mathcal{T}_{k,\tau-1}} \sum_{n \in S_{k,s} \cup \{0\}} y_{n,s} \log p(n|S_{k,s}, \theta)$

 // Assortments Construction

26   $\{S_{l,\tau}^{\alpha,(n,k)}\}_{l \in [K]} \leftarrow \mathbb{O}_{UCB}^{\alpha,(n,k)}$ from (12) for $n \in \mathcal{N}_{k,\tau-1}$ and $k \in [K]$

 // Elimination

27   $\{S_{l,\tau}^\alpha\}_{l \in [K]} \leftarrow \mathbb{O}_{LCB}^\alpha$ from (13)

28   $\mathcal{N}_{k,\tau} \leftarrow \{n \in \mathcal{N}_{k,\tau-1} : \sum_{l \in [K]} \alpha R_{l,\tau-1}^{LCB}(S_l^\alpha) \leq \sum_{l \in [K]} R_{l,\tau-1}^{UCB}(S_{l,\tau}^{\alpha,(n,k)})\}$ for $k \in [K]$

 // G/D-optimal design

29   $\pi_{k,\tau} \leftarrow \arg\max_{\pi \in \mathcal{P}(\mathcal{N}_{k,\tau})} \log\det \sum_{n \in \mathcal{N}_{k,\tau}} \pi_{k,\tau}(n) z_n z_n^\top$ for $k \in [K]$

 // Exploration

30   **for** $k \in [K]$ **do**

31    **for** $n \in \mathcal{N}_{k,\tau}$ **do**

32     $t_{n,k} \leftarrow t, \mathcal{T}_{n,k,\tau}^{(2)} \leftarrow [t_{n,k}, t_{n,k} + \lceil r\pi_{k,\tau}(n)T_\tau \rceil - 1]$

33     **while** $t \in \mathcal{T}_{n,k,\tau}^{(2)}$ **do**

34      Offer $\{S_{k,t}\}_{k \in [K]} = \{S_{l,\tau}^{(n,k)}\}_{l \in [K]}$ and observe feedback $y_{m,t} \in \{0,1\}$ for all $m \in S_{l,t}, l \in [K]$

35      $t \leftarrow t + 1$

36   $T_{\tau+1} \leftarrow 2T_\tau$

---

### A.3.2 PROOF OF THEOREM 2

In this proof, we provide only the parts that are different from the proof of Theorem 1.

**Lemma 10.** *Under E, $(S_1^*, \ldots, S_K^*) \in \mathcal{M}_{\tau-1}$ for all $\tau \in [T]$.*

*Proof.* Here we use induction for the proof. Suppose that for fixed $\tau$, we have $(S_1^*, \ldots, S_K^*) \in \mathcal{M}_\tau$ for all $k \in [K]$. Since $x/(1+x)$ is a non-decreasing function for $x > -1$ and $z_n^\top \hat{\theta}_{k,\tau} - \beta_T \|z_n\|_{V_{k,\tau}^{-1}} \leq z_n^\top \theta_k \leq z_n^\top \hat{\theta}_{k,\tau} + \beta_T \|z_n\|_{V_{k,\tau}^{-1}}$, we have $R_{k,\tau}^{UCB}(S) \geq R_k(S)$ and $R_{k,\tau}^{LCB}(S) \leq$

$R_k(S)$ for any $S \subset [N]$. Then for $k \in [K]$, $n \in S_k^*$, and any $(S_1, .., S_K) \in \mathcal{M}_\tau$, we have

$$
\begin{aligned}
\sum_{l \in [K]} R_{l,\tau}^{UCB}(S_{l,\tau+1}^{\alpha,(n,k)}) &\geq \max_{\{S_k\}_{k \in [K]} \in \mathcal{M}_\tau : n \in S_k} \sum_{l \in [K]} \alpha R_{l,\tau}^{UCB}(S_l) \\
&\geq \sum_{l \in [K]} \alpha R_{l,\tau}^{UCB}(S_l^*) \\
&\geq \sum_{l \in [K]} \alpha R_l(S_l^*) \\
&\geq \sum_{l \in [K]} \alpha R_l(S_{l,\tau}^\alpha) \\
&\geq \sum_{l \in [K]} \alpha R_{l,\tau}^{LCB}(S_{l,\tau}^\alpha),
\end{aligned}
\tag{22}
$$

where the first inequality comes from (12), the second one comes from $(S_1^*, \ldots S_K^*) \in \mathcal{M}_\tau$, and the firth one comes from the optimality of $(S_1^*, \ldots, S_K^*)$. This implies that $n \in \mathcal{N}_{k,\tau+1}$ from the algorithm. Then by following the same statement of (22) for all $n \in S_k^*$ and $k \in [K]$, we have $S_k^* \subset \mathcal{N}_{k,\tau+1}$ for all $k \in [K]$, which implies $(S_1^*, \ldots, S_K^*) \in \mathcal{M}_{\tau+1}$. Therefore, with $(S_1^*, \ldots, S_K^*) \in \mathcal{M}_1$, we can conclude the proof from the induction. $\qquad\square$

From Lemmas 10 and 8, under $E$, we have

$$
\begin{aligned}
\sum_{l \in [K]} \alpha^2 R_l(S_l^*) - \sum_{l \in [K]} R_l(S_{l,\tau}^{\alpha,(n,k)}) &\leq \sum_{l \in [K]} \alpha^2 R_{l,\tau-1}^{LCB}(S_l^*) + 2\beta_T \max_{m \in S_l^*} \|z_m\|_{V_{l,\tau-1}^{-1}} \\
&\quad - \sum_{l \in [K]} R_{l,\tau-1}^{UCB}(S_{l,\tau}^{\alpha,(n,k)}) + 2\beta_T \max_{m \in S_{l,\tau}^{(n,k)}} \|z_m\|_{V_{l,\tau-1}^{-1}} \\
&\leq \sum_{l \in [K]} \alpha R_{l,\tau-1}^{LCB}(S_{l,\tau}^\alpha) + 2\beta_T \max_{m \in S_l^*} \|z_m\|_{V_{l,\tau-1}^{-1}} \\
&\quad - \sum_{l \in [K]} R_{l,\tau-1}^{UCB}(S_{l,\tau}^{\alpha,(n,k)}) + 2\beta_T \max_{m \in S_{l,\tau}^{(n,k)}} \|z_m\|_{V_{l,\tau-1}^{-1}} \\
&\leq 2\beta_T \sum_{l \in [K]} \left( \max_{m \in S_l^*} \|z_m\|_{V_{l,\tau-1}^{-1}} + \max_{m \in S_{l,\tau}^{(n,k)}} \|z_m\|_{V_{l,\tau-1}^{-1}} \right),
\end{aligned}
\tag{23}
$$

where the second inequality comes from (13) and last inequality comes from the fact that $(S_1^*, \ldots, S_K^*) \in \mathcal{M}_{\tau-1}$ and $\sum_{l \in [K]} \alpha R_{l,\tau-1}^{LCB}(S_{l,\tau}^\alpha) \leq \sum_{l \in [K]} R_{l,\tau-1}^{UCB}(S_{l,\tau}^{(n,k)})$ from the algorithm. Then by following the proof in Theorem 1, we can conclude that with $\gamma = \alpha^2$,

$$
\mathcal{R}^\gamma(T) = \widetilde{\mathcal{O}}((1/\kappa)K\sqrt{rKT}).
$$

## A.4 PROOF OF LEMMAS

### A.4.1 PROOF OF LEMMA 2

For the poof, we follow the proof steps in (Bounding the Prediction Error) Oh & Iyengar (2021). We define

$$
H_{k,\tau}(\theta) = \sum_{t \in \mathcal{T}_{k,\tau}} \left( \sum_{n \in S_{k,t}} p(n|S_{k,t}, \theta) z_n z_n^\top - \sum_{n \in S_{k,t}} \sum_{m \in S_{k,t}} p(n|S_{k,t}, \theta) p(m|S_{k,t}, \theta) z_n z_m^\top \right).
$$

We note that $g_{k,\tau}(\theta_1) - g_{k,\tau}(\theta_2) = \sum_{t\in\mathcal{T}_{k,\tau}} \sum_{n\in S_{k,t}}(p(n,|S_{k,t},\theta_1) - p(n,|S_{k,t},\theta_2))z_n$. Then from the mean value theorem, there exists $\bar{\theta} = c\theta_1 + (1-c)\theta_2$ with some $c\in(0,1)$ such that

$$
\begin{aligned}
&g_{k,\tau}(\theta_1) - g_{k,\tau}(\theta_2) \\
&= \nabla_\theta g_{k,\tau}(\theta)\big|_{\theta=\bar{\theta}}(\theta_1 - \theta_2) \\
&= \sum_{t\in\mathcal{T}_{k,\tau}} \left( \sum_{n\in S_{k,t}} p(n|S_{k,t},\bar{\theta})z_n z_n^\top - \sum_{n\in S_{k,t}}\sum_{m\in S_{k,t}} p(n|S_{k,t},\bar{\theta})p(m|S_{k,t},\bar{\theta})z_n z_m^\top \right)(\theta_1 - \theta_2) \\
&= H_{k,\tau}(\bar{\theta})(\theta_1 - \theta_2)
\end{aligned}
\tag{24}
$$

We define $L_{k,\tau} = H_{k,\tau}(\theta_k^*)$ and $E_{k,\tau} = H_{k,\tau}(\bar{\theta}_k) - H_{k,\tau}(\theta_k^*)$ where $\bar{\theta}_k = c\theta_k^* + (1-c)\hat{\theta}_{k,\tau}$ for some constant $c\in(0,1)$.

From (24) we have $g_{k,\tau}(\hat{\theta}_{k,\tau}) - g_{k,\tau}(\theta_k^*) = (L_{k,\tau} + E_{k,\tau})(\hat{\theta}_{k,\tau} - \theta_k^*)$. Then, for any $z\in\mathbb{R}^r$, we have

$$
\begin{aligned}
z^\top(\hat{\theta}_{k,\tau} - \theta_k^*) &= z^\top(L_{k,\tau} + E_{k,\tau})^{-1}(g_{k,\tau}(\hat{\theta}_{k,\tau}) - g_{k,\tau}(\theta_k^*)) \\
&= z^\top L_{k,\tau}^{-1}(g_{k,\tau}(\hat{\theta}_{k,\tau}) - g_{k,\tau}(\theta_k^*)) - z^\top L_{k,\tau}^{-1}E_{k,\tau}(L_{k,\tau} + E_{k,\tau})^{-1}(g_{k,\tau}(\hat{\theta}_{k,\tau}) - g_{k,\tau}(\theta_k^*)).
\end{aligned}
\tag{25}
$$

For obtaining a bound for $|z^\top(\hat{\theta}_{k,\tau} - \theta_k^*)|$, we analyze the two terms in (25). We first provide a bound for $|z^\top L_{k,\tau}^{-1}(g_{k,\tau}(\hat{\theta}_{k,\tau}) - g_{k,\tau}(\theta_k^*))|$. Let $\epsilon_{n,t} = y_{n,t} - p(n|S_{k,t},\theta_k^*)$ for $n\in S_{k,t}$. Since $\hat{\theta}_{k,\tau}$ is the solution from MLE such that $\sum_{t\in\mathcal{T}_{k,\tau}}\sum_{n\in S_{k,t}}(p(n|S_{k,t},\hat{\theta}_{k,\tau}) - y_{n,k,\tau})z_n = 0$, we have

$$
\begin{aligned}
&g_{k,\tau}(\hat{\theta}_{k,\tau}) - g_{k,\tau}(\theta_k^*) \\
&= \sum_{t\in\mathcal{T}_{k,\tau}}\sum_{n\in S_{k,t}} \left( p(n|S_{k,t},\hat{\theta}_{k,\tau}) - p(n|S_{k,t},\theta_k^*) \right) z_n \\
&= \sum_{t\in\mathcal{T}_{k,\tau}}\sum_{n\in S_{k,t}} \left( p(n|S_{k,t},\hat{\theta}_{k,\tau}) - y_{n,k,t} \right) z_n + \sum_{t\in\mathcal{T}_{k,\tau}}\sum_{n\in S_{k,t}} \left( y_{n,k,\tau} - p(n|S_{k,t},\theta_k^*) \right) z_n \\
&= 0 + \sum_{t\in\mathcal{T}_{k,\tau}}\sum_{n\in S_{k,t}} \epsilon_{n,t} z_n
\end{aligned}
\tag{26}
$$

We define

$$
Z_{k,t} = [z_n : n\in S_{k,t}]^\top \in \mathbb{R}^{|S_{k,t}|\times r} \text{ for } t\in\mathcal{T}_{k,\tau},
$$

$$
D_{k,\tau} = [Z_{k,t} : t\in\mathcal{T}_{k,\tau}]^\top \in \mathbb{R}^{(\sum_{t\in\mathcal{T}_{k,\tau}}|S_{k,t}|)\times r},
$$

$$
\mathcal{E}_{k,t} = [\epsilon_{n,t} : n\in S_{k,t}]^\top \in \mathbb{R}^{|S_{k,t}|}.
$$

Then using Hoeffding inequality, we have

$$
\begin{aligned}
\mathbb{P}(|z^\top L_{k,\tau}^{-1}(g_{k,\tau}(\hat{\theta}_{k,\tau}) - g_{k,\tau}(\theta_k^*))| \geq \nu) &= \mathbb{P}\left( \left| \sum_{t\in\mathcal{T}_{k,\tau}} z^\top L_{k,\tau}^{-1} Z_{k,t}^\top \mathcal{E}_{k,t} \right| \geq \nu \right) \\
&\leq 2\exp\left( -\frac{2\nu^2}{\sum_{t\in\mathcal{T}_{k,\tau}}(2\sqrt{2}\|z^\top L_{k,\tau}^{-1} Z_{k,t}^\top\|_2)^2} \right) \\
&= 2\exp\left( -\frac{\nu^2}{4\|z^\top L_{k,\tau}^{-1} D_{k,\tau}^\top\|_2^2} \right) \\
&\leq 2\exp\left( -\frac{\kappa^2\nu^2}{4\|z\|_{V_{k,\tau}^{-1}}^2} \right),
\end{aligned}
\tag{27}
$$

where the last inequality is obtained from the fact that

$$
L_{k,\tau} = \sum_{t \in \mathcal{T}_{k,\tau}} \left( \sum_{n \in S_{k,t}} p(n|S_{k,t}, \theta_k^*) z_n z_n^\top - \sum_{n \in S_{k,t}} \sum_{m \in S_{k,t}} p(n|S_{k,t}, \theta_k^*) p(m|S_{k,t}, \theta_k^*) z_n z_m^\top \right)
$$

$$
= \sum_{t \in \mathcal{T}_{k,\tau}} \left( \sum_{n \in S_{k,t}} p(n|S_{k,t}, \theta_k^*) z_n z_n^\top - \frac{1}{2} \sum_{n \in S_{k,t}} \sum_{m \in S_{k,t}} p(n|S_{k,t}, \theta_k^*) p(m|S_{k,t}, \theta_k^*) (z_n z_m^\top + z_m z_n^\top) \right)
$$

$$
\succeq \sum_{t \in \mathcal{T}_{k,\tau}} \left( \sum_{n \in S_{k,t}} p(n|S_{k,t}, \theta_k^*) z_n z_n^\top - \frac{1}{2} \sum_{n \in S_{k,t}} \sum_{m \in S_{k,t}} p(n|S_{k,t}, \theta_k^*) p(m|S_{k,t}, \theta_k^*) (z_n z_n^\top + z_m z_m^\top) \right)
$$

$$
= \sum_{t \in \mathcal{T}_{k,\tau}} \left( \sum_{n \in S_{k,t}} p(n|S_{k,t}, \theta_k^*) z_n z_n^\top - \sum_{n \in S_{k,t}} \sum_{m \in S_{k,t}} p(n|S_{k,t}, \theta_k^*) p(m|S_{k,t}, \theta_k^*) z_n z_n^\top \right)
$$

$$
= \sum_{t \in \mathcal{T}_{k,\tau}} \left( \sum_{n \in S_{k,t}} p(n|S_{k,t}, \theta_k^*) p(n_0|S_{k,t}, \theta_k^*) z_n z_n^\top \right) \succeq \kappa D_\tau^\top D_\tau (= \kappa V_{k,\tau}),
$$

where the first inequality is obtained from $(z_n - z_m)(z_n - z_m)^\top = z_n z_n^\top + z_m z_m^\top - z_n z_m^\top - z_m z_n^\top \succeq 0$.

Then from (27) using $\nu = (1/\kappa)\sqrt{\log(2TKN/\delta)}\|z\|_{V_{k,\tau}^{-1}}$ and the union bound, with probability at least $1 - \delta$, for all $\tau \in [T], k \in [K]$, we have

$$
|z^\top L_{k,\tau}^{-1}(g_{k,\tau}(\hat{\theta}_{k,\tau}) - g_{k,\tau}(\theta_k^*))| \leq \frac{2\sqrt{\log(TKN/\delta)}}{\kappa}\|z\|_{V_{k,\tau}^{-1}}. \tag{28}
$$

Now we provide a bound for the second term in (25) of $|z^\top L_{k,\tau}^{-1} E_{k,\tau}(L_{k,\tau} + E_{k,\tau})^{-1}(g_{k,\tau}(\hat{\theta}_{k,\tau}) - g_{k,\tau}(\theta_k^*))|$. We have

$$
|z^\top L_{k,\tau}^{-1} E_{k,\tau}(L_{k,\tau} + E_{k,\tau})^{-1}(g_{k,\tau}(\hat{\theta}_{k,\tau}) - g_{k,\tau}(\theta_k^*))|
$$

$$
\leq \|z\|_{L_{k,\tau}^{-1}} \|L_{k,\tau}^{-1/2} E_{k,\tau}(L_{k,\tau} + E_{k,\tau})^{-1} L^{1/2}\|_2 \|g_{k,\tau}(\hat{\theta}_{k,\tau}) - g_{k,\tau}(\theta_k^*)\|_{L_{k,\tau}^{-1}}
$$

$$
\leq (1/\kappa)\|z\|_{V_{k,\tau}^{-1}} \|L_{k,\tau}^{-1/2} E_{k,\tau}(L_{k,\tau} + E_{k,\tau})^{-1} L^{1/2}\|_2 \|g_{k,\tau}(\hat{\theta}_{k,\tau}) - g_{k,\tau}(\theta_k^*)\|_{V_{k,\tau}^{-1}}. \tag{29}
$$

Then it follows that

$$
\|L_{k,\tau}^{-1/2} E_{k,\tau}(L_{k,\tau} + E_{k,\tau})^{-1} L^{1/2}\|_2
$$

$$
= \|L_{k,\tau}^{-1/2} E_{k,\tau}(L_{k,\tau}^{-1} - L_{k,\tau}^{-1} E_{k,\tau}(L_{k,\tau} + E_{k,\tau})^{-1} L^{1/2}\|_2
$$

$$
\leq \|L_{k,\tau}^{-1/2} E_{k,\tau} L_{k,\tau}^{-1/2}\|_2 + \|L_{k,\tau}^{-1/2} E_{k,\tau} L_{k,\tau}^{-1/2}\|_2 \|L_{k,\tau}^{-1/2} E_{k,\tau}(L_{k,\tau} + E_{k,\tau})^{-1} L_{k,\tau}^{1/2}\|_2,
$$

which implies

$$
\|L_{k,\tau}^{-1/2} E_{k,\tau}(L_{k,\tau} + E_{k,\tau})^{-1} L_{k,\tau}^{1/2}\|_2 \leq \frac{\|L_{k,\tau}^{-1/2} E_{k,\tau} L_{k,\tau}^{-1/2}\|_2}{1 - \|L_{k,\tau}^{-1/2} E_{k,\tau} L_{k,\tau}^{-1/2}\|_2}
$$

$$
\leq 2\|L_{k,\tau}^{-1/2} E_{k,\tau} L_{k,\tau}^{-1/2}\|_2
$$

$$
\leq \frac{6}{\kappa}\|\hat{\theta}_{k,\tau} - \theta_k^*\|_2, \tag{30}
$$

where the last inequality is obtained from (17) and (18) in Oh & Iyengar (2021). Then from (29), (30), we have

$$
|z^\top L_{k,\tau}^{-1} E_{k,\tau}(L_{k,\tau} + E_{k,\tau})^{-1}(g_{k,\tau}(\hat{\theta}_{k,\tau}) - g_{k,\tau}(\theta_k^*))|
$$

$$
\leq \frac{6}{\kappa^2}\|\hat{\theta}_{k,\tau} - \theta_k^*\|_2 \|g_{k,\tau}(\hat{\theta}_{k,\tau}) - g_{k,\tau}(\theta_k^*)\|_{V_{k,\tau}^{-1}} \|z\|_{V_{k,\tau}^{-1}}. \tag{31}
$$

We can conclude the proof from (28) and (31).

### A.4.2 PROOF OF LEMMA 6

We note that $g_{k,\tau}(\theta_1) - g_{k,\tau}(\theta_2) = \sum_{t \in \mathcal{T}_{k,\tau}} \sum_{n \in S_{k,t}} (p(n, |S_{k,t}, \theta_1) - p(n, |S_{k,t}, \theta_2))z_n$. Define
$H_{k,\tau}(\theta) = \sum_{t \in \mathcal{T}_{k,\tau}} \left( \sum_{n \in S_{k,t}} p(n|S_{k,t}, \theta)z_n z_n^\top - \sum_{n \in S_{k,t}} \sum_{m \in S_{k,t}} p(n|S_{k,t}, \theta)p(m|S_{k,t}, \theta)z_n z_m^\top \right)$.
Then we can show that there exists $\bar{\theta} = c\theta_1 + (1-c)\theta_2$ with some $c \in (0,1)$ such that

$$g_{k,\tau}(\theta_1) - g_{k,\tau}(\theta_2)$$
$$= \nabla_\theta g_{k,\tau}(\theta)\big|_{\theta=\bar{\theta}}(\theta_1 - \theta_2)$$

$$= \sum_{t \in \mathcal{T}_{k,\tau}} \left( \sum_{n \in S_{k,t}} p(n|S_{k,t}, \bar{\theta})z_n z_n^\top - \sum_{n \in S_{k,t}} \sum_{m \in S_{k,t}} p(n|S_{k,t}, \bar{\theta})p(m|S_{k,t}, \bar{\theta})z_n z_m^\top \right)(\theta_1 - \theta_2)$$

$$= H_{k,\tau}(\bar{\theta})(\theta_1 - \theta_2). \tag{32}$$

Define $\bar{H}_{k,\tau}(\bar{\theta}) = \sum_{t \in \mathcal{T}_{k,\tau}} \sum_{n \in S_{k,t}} p(n|S_{k,t}, \bar{\theta})p(n_0|S_{k,t}, \bar{\theta})z_n z_n^\top$. Then we have $H_{k,\tau}(\bar{\theta}) \succeq \bar{H}_{k,\tau}(\bar{\theta})$ from the following.

$$\sum_{t \in \mathcal{T}_{k,\tau}} \left( \sum_{n \in S_{k,t}} p(n|S_{k,t}, \bar{\theta})z_n z_n^\top - \sum_{n \in S_{k,t}} \sum_{m \in S_{k,t}} p(n|S_{k,t}, \bar{\theta})p(m|S_{k,t}, \bar{\theta})z_n z_m^\top \right)$$

$$= \sum_{t \in \mathcal{T}_{k,\tau}} \left( \sum_{n \in S_{k,t}} p(n|S_{k,t}, \bar{\theta})z_n z_n^\top - \sum_{n \in S_{k,t}} \sum_{m \in S_{k,t}} p(n|S_{k,t}, \bar{\theta})p(m|S_{k,t}, \bar{\theta})z_n z_m^\top \right)$$

$$= \sum_{t \in \mathcal{T}_{k,\tau}} \left( \sum_{n \in S_{k,t}} p(n|S_{k,t}, \bar{\theta})z_n z_n^\top - \frac{1}{2} \sum_{n \in S_{k,t}} \sum_{m \in S_{k,t}} p(n|S_{k,t}, \bar{\theta})p(m|S_{k,t}, \bar{\theta})(z_n z_m^\top + z_m z_n^\top) \right)$$

$$\succeq \sum_{t \in \mathcal{T}_{k,\tau}} \left( \sum_{n \in S_{k,t}} p(n|S_{k,t}, \bar{\theta})z_n z_n^\top - \frac{1}{2} \sum_{n \in S_{k,t}} \sum_{m \in S_{k,t}} p(n|S_{k,t}, \bar{\theta})p(m|S_{k,t}, \bar{\theta})(z_n z_n^\top + z_m z_m^\top) \right)$$

$$= \sum_{t \in \mathcal{T}_{k,\tau}} \left( \sum_{n \in S_{k,t}} p(n|S_{k,t}, \bar{\theta})z_n z_n^\top - \sum_{n \in S_{k,t}} \sum_{m \in S_{k,t}} p(n|S_{k,t}, \bar{\theta})p(m|S_{k,t}, \bar{\theta})z_n z_n^\top \right)$$

$$= \sum_{t \in \mathcal{T}_{k,\tau}} \left( \sum_{n \in S_{k,t}} p(n|S_{k,t}, \bar{\theta})p(n_0|S_{k,t}, \bar{\theta})z_n z_n^\top \right), \tag{33}$$

where the inequality is obtained from $(z_n - z_m)(z_n - z_m)^\top \succeq 0$. Under $E_1$, we have $\|\hat{\theta}_{k,\tau}\|_2 - \|\theta_k^*\|_2 \leq 1$ implying $\|\hat{\theta}_{k,\tau}\|_2 \leq 1 + \|\theta_k^*\|_2 = 1 + \|U_r^\top \theta_k\|_2 \leq 2$. Then for $\bar{\theta} = c\hat{\theta}_{k,\tau} + (1 - c)\theta_k^*$ for some $c \in (0,1)$, we have $\|U_r \bar{\theta}\|_2 \leq 2$. Then from Assumption 2 and $p(n|S_{k,t}, \bar{\theta}) = \exp(z_n^\top \bar{\theta})/(1 + \sum_{m \in S_{k,t}} \exp(z_m^\top \bar{\theta})) = \exp(x_n^\top (U_r \bar{\theta}))/(1 + \sum_{m \in S_{k,t}} \exp(x_m^\top (U_r \bar{\theta})))$, we can show that $\bar{H}_{k,\tau}(\bar{\theta}) \succeq \kappa V_{k,\tau}$, which implies $H_{k,\tau}(\bar{\theta}) \succeq \bar{H}_{k,\tau}(\bar{\theta}) \succeq \kappa V_{k,\tau}$.

Then we have

$$\|\hat{\theta}_{k,\tau} - \theta_k^*\|_2^2 \leq (1/\lambda_{\min}(V_{k,\tau}))(\hat{\theta}_{k,\tau} - \theta_k^*)^\top V_{k,\tau}(\hat{\theta}_{k,\tau} - \theta_k^*)$$
$$\leq (1/\kappa\lambda_{\min}(V_{k,\tau}^0))(\hat{\theta}_{k,\tau} - \theta_k^*)^\top H_{k,\tau}(\bar{\theta})(\hat{\theta}_{k,\tau} - \theta_k^*)$$
$$\leq (1/\kappa\lambda_{\min}(V_{k,\tau}^0))(\hat{\theta}_{k,\tau} - \theta_k^*)^\top H_{k,\tau}(\bar{\theta})H_{k,\tau}(\bar{\theta})^{-1}H_{k,\tau}(\bar{\theta})(\hat{\theta}_{k,\tau} - \theta_k^*)$$
$$\leq (1/\kappa^2\lambda_{\min}(V_{k,\tau}^0))(g_{k,\tau}(\hat{\theta}_{k,\tau}) - g_{k,\tau}(\theta_k^*))^\top V_{k,\tau}^{-1}(g_{k,\tau}(\hat{\theta}_{k,\tau}) - g_{k,\tau}(\theta_k^*))$$
$$\leq (1/\kappa^2\lambda_{\min}(V_{k,\tau}^0))\|g_{k,\tau}(\hat{\theta}_{k,\tau}) - g_{k,\tau}(\theta_k^*)\|_{V_{k,\tau}^{-1}}^2. \tag{34}$$

Then from $E_2$, we can conclude that

$$\|\hat{\theta}_{k,\tau} - \theta_k^*\|_2 \leq \frac{4}{\kappa}\sqrt{\frac{2r + \log(KTN/\delta)}{\lambda_{\min}(V_{k,\tau}^0)}}.$$

## A.5 Additional Experiments for Gaussian Distribution for features

We conduct experiments using features drawn from a Gaussian distribution with a mean of zero and a variance of one. After generation, the features are normalized. The results are presented in Figure 3.

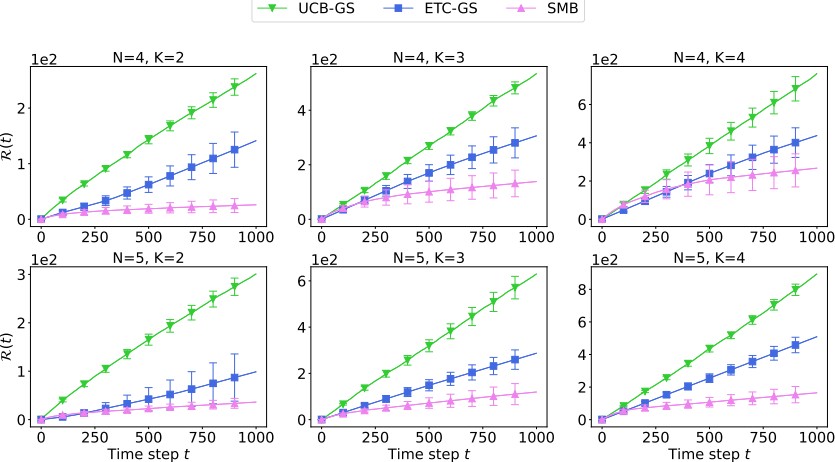

Figure 3: Experimental results for regret of algorithms under Gaussian distribution for features

