# OpenReview forum: "Stochastic Matching Bandits under Preference Feedback"
_ICLR.cc/2025/Conference — ICLR 2025 Conference Withdrawn Submission_

### Official Review · Reviewer_cpX4 · 2024-10-22

**Soundness:** 2
**Presentation:** 3
**Contribution:** 2
**Rating:** 5
**Confidence:** 3

**Summary:**

This paper proposes a new bandit framework of stochastic matching model, where agents on one side are assigned to arms on the other side, and each arm stochastically accepts an agent among the assigned pool of agents based on its unknown preference, allowing a possible outside option of not accepting any.

**Strengths:**

- This paper proposes a new bandit framework of matching, where the agents can be stochastically rejected from the arm side. This framework has some reasonable real-world applications.
- It discusses the computation issue of the algorithm, which often becomes one of the main aspects of combinatorial badits. It shows a regret bound when using an $\alpha$-approximation oracle.

**Weaknesses:**

- There is no lower bound for this framework, which weakens the contribution since we can not assess the upper bound quantitatively. Won't Merlis et al. (2020) help analyze the lower bound of MNL?
- Although it proposes an algorithm with an approximation oracle, it is not shown in the experiment. It would be even better if there were numerical experiments showing that using an $\alpha$-approximation oracle improves calculation execution time.



[Reference]
- N. Merlis et al. Tight Lower Bounds for Combinatorial Multi-Armed Bandits. ACLT2020.

**Questions:**

- On page 7, there is a proof sketch for Theorem 1. Is this necessary? If so, could you briefly explain which part of the proof is novel?
- $\kappa$ seems to be the only parameter related to the rejection of the arm side. Thinking straightforwardly, can't we directly define like $(\kappa_{ij})\_{i = 1, ..., n, j = 1, ..., k}$, where $\kappa_{ij}$ is the probability that agent $i$ will be rejected from arm $j$?
- S. Wang (ICML2018) should be cited since this work also works on combinatorial bandit with semi-bandit feedback. Shouldn't their algorithm, CTS, at least be compared in the experiment section?
- I would like to know if there is a proper real-world dataset for the experiment. This study would be even more interesting to have a numerical experiment with real-world data since it seems to be more focused on real-world applications.

[Reference]
- Siwei Wang and Wei Chen. Thompson Sampling for Combinatorial Semi-Bandits. ICML, 2018.

---

> ### Author Response · Authors · 2024-11-20
>
> We appreciate your effort in reviewing our paper and providing insightful and valuable feedback. Below, we have addressed each of your comments and questions.
>
>
> $\bullet$ **There is no lower bound for this framework, which weakens the contribution since we can not assess the upper bound quantitatively. Won't Merlis et al. (2020) help analyze the lower bound of MNL?**
>
>
> **Response:** The primary challenge in deriving a regret lower bound for our framework stems from its fundamental differences from standard combinatorial multi-armed bandit (CMAB) settings. In our problem, we consider preference feedback under the MNL structure, where the expected reward (or preference) for each arm depends on the assigned assortment of agents, rather than being fixed as in typical CMAB problems. While Merlis et al. (2020) provide insights into the lower bounds for MNL in certain settings, their focus does not fully align with our multi-agent, preference-based matching framework. Extending their techniques or adapting them to our context would require significant modifications and is not straightforward.
>
> Our regret bounds are **tight with respect to $T$ and $r$**, as they exhibit the expected $\sqrt{rT}$-style growth commonly observed in structured bandit literature. However, the dependency on $K\sqrt{K}$ in our regret bounds reflects the combinatorial complexity of matching multiple agents to arms with $K$ latent parameters. Since the regret accounts for successes across $K$ different arms with $K$ latent parameters, we conjecture that achieving better than a linear dependence on $K$ in the regret is unlikely, implying that the regret lower bound lies between $K$ and $K\sqrt{K}$. Achieving tighter regret bounds with respect to $K$ remains an open challenge.
>
> $\bullet$ **On page 7, there is a proof sketch for Theorem 1. Is this necessary? If so, could you briefly explain which part of the proof is novel?**
>
>  **Response:** Naive approach for exploring all possible matching candidates occurs high cost. It is novel how to explore all possible matching under statistical efficiency. We propose to explore each assignment $(n,k)$ by constructing a representative assortment $\{S_{l,\tau}^{(n,k)}\}$ in Eq. (3). Under this exploration process and elimination, it is challenging to show the guarantee that the optimal assortment is included in the active matching set, $\mathcal{M}_{\tau-1}$, which is shown in Eqs. (5) and (6) using induction.
>  We also highlight this is the first work for proposing algorithms using G/D-optimal design in matching bandits under preference feedback.
>
>
>
> $\bullet$   **$\kappa$ seems to be the only parameter related to the rejection of the arm side. Thinking straightforwardly, can't we directly define like $\kappa_{i,j}$, where
>  $\kappa_{i,j}$ is the probability that agent $i$ will be rejected from arm $j$?**
>
>  **Response:** As highlighted in previous MNL/logistic bandit literature [1,2], $\kappa$ is closely tied to the **non-linearity** of the problem. A smaller $\kappa$ value reflects greater non-linearity, which increases the difficulty of learning in this setting.
>
> Introducing $\kappa_{i,j}$ instead of a single global $\kappa$ does not add significant value. While it may be possible to define such parameters, we do not believe this would lead to different regret bounds. The regret bound would depend on the specific values of $\kappa_{i,j}$ for the assigned assortment at each time step. Over the entire time horizon and all possible assortments, the regret would ultimately rely on the **worst-case value** of $\kappa_{i,j}$ across all assignments and time steps.
>
> In our current analysis, $\kappa$ is defined as a global parameter that provides a uniform lower bound across all assortments. If the regret bound is determined by the worst-case of $\kappa_{i,j}$, this simplifies to $\kappa = \min_{i,j} \kappa_{i,j}$, which is functionally equivalent to the existing definition of $\kappa$. Therefore, introducing $\kappa_{i,j}$ does not fundamentally change the regret analysis or its interpretation.
>
>
> [1]Faury, Louis, et al. "Improved optimistic algorithms for logistic bandits." International Conference on Machine Learning. PMLR, 2020.
>
> [2] Goyal, Vineet, and Noemie Perivier. "Dynamic pricing and assortment under a contextual mnl demand." arXiv preprint arXiv:2110.10018 (2021).

---

> ### Author Response · Authors · 2024-11-20
>
> $\bullet$ **S. Wang (ICML2018) should be cited since this work also works on combinatorial bandit with semi-bandit feedback. Shouldn't their algorithm, CTS, at least be compared in the experiment section?**
>
> **Response:** We appreciate the suggestion to reference S. Wang's work (ICML 2018) on combinatorial bandits with semi-bandit feedback. We will add the following discussion in our final,  along with a reference to  S. Wang's work (ICML 2018).
>
>
> For the experiment, however, there are fundamental differences between our framework and the combinatorial multi-armed bandit (CMAB) setting explored in that work, making a direct comparison less meaningful.
>
>
>
>
> Our framework introduces preference feedback where the expected reward for each arm is not fixed but instead depends on the assigned assortment of agents. This contrasts with CMAB, where each reward of an arm is typically independent of other assigned arms. In the MNL model, an arm's preference for an agent is relative to the other agents in its assigned assortment, introducing dependencies across the chosen agents.
>
> Furthermore, our setting involves matching multiple agents to multiple arms simultaneously. The reward (or successful match) depends on the stochastic acceptance behavior of the arms based on their preferences, which are governed by the MNL model. Extending CMAB frameworks, such as CTS, to handle this multi-agent and multi-arm setting is non-trivial and would require significant modifications.
>
>
> While S. Wang’s work provides valuable insights into semi-bandit feedback, it does not align with the key aspects of our framework, such as MNL-based preference feedback and the multi-agent matching problem. For these reasons, comparing with CTS in the experimental section would not accurately reflect the contributions or performance of our algorithm.
>
>
>
> $\bullet$ **I would like to know if there is a proper real-world dataset for the experiment. This study would be even more interesting to have a numerical experiment with real-world data since it seems to be more focused on real-world applications.**
>
> **Response:**  Please note that conducting experiments in bandit settings using real-world offline datasets is inherently challenging due to the nature of partial feedback in online learning.
> Bandit algorithms rely on feedback that is contingent upon the actions they take, whereas offline datasets typically lack counterfactual feedback—i.e., the outcomes of actions not taken during data collection. This fundamental limitation makes it difficult to directly evaluate bandit algorithms on fixed real-world datasets without significant adjustments.
>
> If any, offline datasets must be transformed into semi-synthetic simulations, where missing counterfactual feedback is either imputed or simulated based on assumptions.
> This transformation effectively reduces real-world data to synthetic, analogous synthetic simulations that we performed in our paper.
>
> For these reasons, we adopted the approach commonly used in the bandit literature, focusing on theoretical results validated with synthetic datasets. This methodology allows for controlled experimentation and ensures that the theoretical properties of the algorithm are demonstrated under well-defined conditions. However, if the reviewer requests and points to a particular dataset, we are open to performing additional experiments using a semi-synthetic approach with a real-world dataset.
>
> $\bullet$ **Although it proposes an algorithm with an approximation oracle, it is not shown in the experiment.**
>
> **Response:** We present a general framework that utilizes an approximation oracle to address computational challenges in combinatorial optimization. The benefit of this approach regarding computation is evident when an appropriate oracle is selected (e.g. Kapralov et al. (2013)). If needed, we are happy to include experimental validation in the final version.

---

> > ### Author Response · Authors · 2024-11-29
> >
> > Thank you again for your valuable feedback and comments. We hope we have addressed your questions and concerns. As the deadline is approaching, please let us know if you have any further questions or comments.

---

> > > ### Comment · Reviewer_cpX4 · 2024-11-29
> > >
> > > Thank you, I have no more questions. I will examine the comments I received in more detail.
> > > For now, I will leave the score as it is.

---

### Official Review · Reviewer_fTET · 2024-10-24

**Soundness:** 2
**Presentation:** 3
**Contribution:** 2
**Rating:** 3
**Confidence:** 4

**Summary:**

The aim of this paper is to study a stochastic matching problem with multinomial logit choices in selecting agents.

The authors consider a model that has N agents with K arms. Each agent n as a d-dimensional feature vector, while each arm k has a d-dimensional latent vector \theta_k.  In each period, the algorithm assign some agents to an arm.  Each arm will probabilistically select an agent (including a null agent which implies no selection of any assigned agent) which follows the Multi-nomial Logit (MNL) model.  The goal is to maximize the expected number of successful matching.

The authors extend the previous works in two fronts:
   - allows select the null-agent (or not selecting any of the assigned agent);
   - instead of having a "deterministic preference" of accepting agents, the authors consider a "probabilistic model" using the multinomial logit function.

The authors proposed an elimination-based algorithm that wherein the regret bound is O(ln(T)) (plus other terms).  Also, there is a computational issues in their proposed elimination-based algorithm,  To address this, the authors an approximation (or \alpha-approximation oracle).

Finally, the authors carried out simulation to illustrate the merits of their works.

**Strengths:**

The strength in this paper, as far as I am concern, is in the writing.

The authors did a good job in presenting the previous and related works, as well as in presenting the mathematical models.

The proposed algorithm (e.g., elimination-based) as well as the alpha-approximation algorithm, as far as I am concerned, are really simple extension of some of the previous algorithms.

The technical proof looks correct, but then again, if one is familiar about MAB or matching MAB, many of the previous works also followed the same line/style in proving the regret bounds.

**Weaknesses:**

The major concern I have about this paper is about its novelty.

First of all, extending from previous matching MAB problem by allowing a null-agent (or arm can reject any of the assigned agent) is really a trivial extension.  In fact, many previous papers can  easily incorporate this case in their model.

Secondly, allowing arms to select an agent based on multinomial logit function for me is in fact a restriction.  Because in real-life, the preference probability can in fact be more "general" then the multinomial logit representation.  One may have a hard time fitting an arbitrary probability distribution using the multinomial logit function.

Thirdly, the elimination-based algorithm is very similar to the previous work by Lattimore and Szepesvari (with minor alteration due to the mapping with the problem).  To resolve the computational issue using an approximation algorithm is also a very common technique, as stated by the authors, from Kakade et at in 2007. In fact, this form of alpha-approximation has been heavily used in many disciplines, from theoretical computer science to machine learning.

Fourthly, the explanation of the algorithm has some issues. The authors discussed about using SVD to compute X=U\sumV^T, but are they defined ?  It seems the authors just use the previous SVD approach to extract information.

Lastly, why just compare with ETC-GS and  UCB-GS?  For example, I want to see how the algorithm will compare when the preference is "deterministic" (as used by many previous matching MAB algorithms), and to see how the variation from the deterministic preference to stochastic preference can impact the complexity and accuracy of the proposed algorithms.

**Questions:**

- Explain and define X=U \sum V^T.  How you get these from the inputs to your algorithm.

- Why multinomial logit is not a restriction?  Since one needs to find the right values of \theta so as to match to any probabilitic distribution.

- Can your algorithm handles time-varying preference in selecting agents?

- Authors need to do a more comprehensive benchmarking and evaluation by comparing with other MAB-matching works.

---

> ### Author Response · Authors · 2024-11-20
>
> We appreciate your effort in reviewing our paper and providing valuable feedback. Below, we have addressed each of your comments and questions. We sincerely hope that we can clarify our contributions.
>
> $\bullet$ **First of all, extending from previous matching MAB problem by allowing a null-agent (or arm can reject any of the assigned agent) is really a trivial extension. In fact, many previous papers can easily incorporate this case in their model.**
>
>  **Response:** The main contribution of our work is to introduce **stochastic matching process** in the matching bandits under preference feedback. Unlike prior work assuming deterministic arm behavior, our framework models stochastic decisions with unknown preferences, aligning with real-world scenarios like online labor markets. To the best of our knowledge, no previous work addresses this setting, making our approach fundamentally novel.
>
> $\bullet$ **Secondly, allowing arms to select an agent based on multinomial logit function for me is in fact a restriction. Because in real-life, the preference probability can in fact be more "general" then the multinomial logit representation. One may have a hard time fitting an arbitrary probability distribution using the multinomial logit function.**
>
>  **Response:** Most prior works on bandits with binary preferences, such as those following the Bradley-Terry model [1,2], restrict feedback to pairwise comparisons. Our model generalizes this by allowing arms to make multiple-choice decisions, capturing richer interactions in matching scenarios.
>
> Moreover, the MNL model has been widely adopted in the bandit community for multi-choice preference feedback due to its mathematical tractability and ability to capture probabilistic preferences [Agrawal et al. (2017b); Chen et al. (2023); Oh \& Iyengar (2019; 2021)]. The MNL model provides a practical and widely recognized framework for modeling preferences in real-world applications such as online marketplaces and recommendation systems.
>
> We acknowledge that any parametric model, including the multinomial logit (MNL), inherently involves approximation. However, this is a standard approach across bandit and online learning problems. For instance, widely used models such as linear or logistic reward functions in bandit settings approximate the underlying reward function. These approximations are not considered limitations but rather practical tools to balance expressiveness and computational tractability.
>
> Similarly, the MNL model provides a structured way to represent probabilistic preferences. Learning $\theta$, the latent parameter governing preferences, is central to leveraging the MNL framework. This process is not a limitation but rather a typical learning objective that ensures model interpretability and alignment with real-world applications.
>
>
> We believe this balance between generality and tractability makes the MNL model a suitable choice for our framework.
>
> [1] Bengs, Viktor, et al. "Preference-based online learning with dueling bandits: A survey." Journal of Machine Learning Research 22.7 (2021): 1-108.
> [2] Szörényi, Balázs, et al. "Online rank elicitation for plackett-luce: A dueling bandits approach." Advances in neural information processing systems 28 (2015).

---

> ### Author Response · Authors · 2024-11-20
>
> $\bullet$ **Thirdly, the elimination-based algorithm is very similar to the previous work by Lattimore and Szepesvari (with minor alteration due to the mapping with the problem). To resolve the computational issue using an approximation algorithm is also a very common technique, as stated by the authors, from Kakade et at in 2007. In fact, this form of alpha-approximation has been heavily used in many disciplines, from theoretical computer science to machine learning.**
>
> **Response:** While we adopt the G/D-optimal design approach from Lattimore and Szepesvári, our contribution goes beyond a direct application of their method. The key difference lies in **how we explore possible matchings** in the stochastic matching bandit problem. A naive approach that explores all possible matching candidates incurs prohibitive computational costs. Instead, we explore at the level of individual edges, which significantly reduces computational complexity.  We proposed a statistically efficient exploration strategy by constructing representative assortments $\{S_{l,\tau}^{(n,k)}\}$ (Eq. 3). However, this introduces a challenge: ensuring that the **optimal assortment** remains in the active matching set $\mathcal{M}_{\tau-1}$. Addressing this challenge requires careful analysis, as detailed in **Lemma 7*, where we prove the inclusion of the optimal assortment in the active set.
>
> Additionally, while the prior work focuses on linear bandits, our framework deals with the more complex MNL model under preference feedback. The confidence bound derivation in our setting differs from the linear bandit case and is specifically tailored to accommodate the probabilistic nature of the MNL model.
>
> These contributions reflect meaningful advances over prior works, tailored to the unique challenges of our problem setting.
>
>
>
> $\bullet$ **Lastly, why just compare with ETC-GS and UCB-GS? For example, I want to see how the algorithm will compare when the preference is "deterministic" (as used by many previous matching MAB algorithms), and to see how the variation from the deterministic preference to stochastic preference can impact the complexity and accuracy of the proposed algorithms.**
>
> **Response:** As noted in lines 469–471, no benchmark exists for the stochastic matching bandit scenario, so we compare our method with standard baselines like UCB-GS and ETC-GS. We believe conducting experiments under a deterministic preference model is not meaningful for two reasons:
>
>
> (Real-World Relevance) Deterministic arm behavior does not align with many real-world scenarios, such as online labor markets or ride-hailing services, where decisions by arms (e.g., laborers or drivers) exhibit stochasticity in behavior from preferences. Our framework addresses this more realistic setting, which goes beyond the deterministic assumptions of earlier works.
>
> (Learning Preferences) In a deterministic preference model, it is impossible to estimate the preference utility between arms and agents because the deterministic outcome does not provide feedback on the strength of preferences. For instance, in deterministic settings, one cannot infer "how much" an arm prefers one agent over others, limiting the richness of the learned model and its applicability to realistic matching problems.
>
> For those reasons, our work focuses on stochastic matching bandits rather than deterministic preference.
>
>
>
> $\bullet$ **Explain and define $X=U \sum V^T$. How you get these from the inputs to your algorithm. (Fourthly, the authors discussed about using SVD to compute X=U\sumV^T, but are they defined ?)**
>
> **Response:** The contribution of our work does not lie in proposing a new SVD method. Instead, we leverage existing SVD techniques to enhance performance in terms of dimensionality reduction in our algorithm.  The SVD can be computed using the following methods.
> (Exact Methods) Classical SVD algorithms, as described in [1], ensure an accurate decomposition of $X$.
> (Iterative Approximation Methods) Approaches like the Power Method [2] can be used for large matrices where computational efficiency is essential. These methods approximate singular vectors iteratively, trading off some accuracy for speed.
>
>
> [1]Dembele, Doulaye. "A Power Method for Computing Singular Value Decomposition." arXiv preprint arXiv:2410.23999 (2024).
>
> [2] Golub, G. H., \& Van Loan, C. F. (2013). "Matrix Computations" (4th ed.).
>
>
> $\bullet$ **Can your algorithm handles time-varying preference in selecting agents?**
>
> **Response:**  Our current problem formulation and algorithm are designed specifically for a static setting, where the preferences of arms over agents remain constant over time. This assumption allows us to focus on the stochastic nature of matching and preference feedback in the static case. While our current work focuses on the static case, extending the framework to handle time-varying preferences is an exciting and challenging direction for future research.

---

> > ### Author Response · Authors · 2024-11-20
> >
> > $\bullet$ **Authors need to do a more comprehensive benchmarking and evaluation by comparing with other MAB-matching works.**
> >
> >
> > **Response:** As noted in lines 469–471, no dedicated benchmark currently exists for our proposed stochastic matching bandit framework, making it challenging to directly compare with other works in this specific setting. This is why we rely on standard matching bandit algorithms such as UCB-GS and ETC-GS for benchmarking, as they are commonly used in the field.
> >
> > We strongly believe that existing matching bandit algorithms are not working properly due to fundamental differences in the problem setting. Most prior works assume deterministic behavior of arms based on known or static preferences, which does not align with our stochastic setting where arms exhibit probabilistic acceptance based on unknown preferences. The stochastic nature of our framework significantly alters the dynamics of learning and decision-making, making it unlikely that existing methods would perform effectively in this context.
> >
> > While we acknowledge the importance of comprehensive benchmarking, it is difficult to conduct fair comparisons when the objectives and underlying assumptions of other algorithms differ significantly from our setting.  We hope this work inspires further research on stochastic matching bandits.

---

> > > ### Author Response · Authors · 2024-11-29
> > >
> > > Thank you again for your valuable feedback and comments. We hope we have addressed your questions and concerns. As the deadline is approaching, please let us know if you have any further questions or comments.

---

### Official Review · Reviewer_ZFHR · 2024-10-29

**Soundness:** 3
**Presentation:** 3
**Contribution:** 2
**Rating:** 6
**Confidence:** 4

**Summary:**

This paper studies a new kind of matching bandit model, in which several agents are assigned to an arm, and the arm accepts one of them (or none of them) following an MNL setting. The goal is to maximize the matching probability. Under general linear structure, the authors propose an elimination-based algorithm called SMB, which achieves a regret upper bound of $\tilde{O}(\sqrt{dK^3T}/\kappa )$. They also propose an approximate-oracle version algorithm to achieve a similar approximate regret, and use experiments to demonstrate the effectiveness of their algorithm.

**Strengths:**

The new model setting is well-motivated.

I read some of the proofs and they seem to be correct.

The writting is easy to understand (though there are some typos).

**Weaknesses:**

I think the primary contribution of this paper is proposing the new model setting. The contributions from the perspective of theoretical analysis seem limited, as they all follow the existing frameworks.

The SMB algorithm is not efficient, and its approximate version needs to use $\alpha$-oracle, but can only obtain an sublinear $\alpha^2$-approxiamte regret upper bound.

Since the target and dynamics of the model is totally different with UCB-GS and ETC-GS, it is not really fair to compare with them.


Some minor typos:

- In Assumption 2, should it be $\inf_{||\theta||_2 \le 1}$?
- In Eq. (4), it should be $R^{LCB}$ but not $R^{LBC}$.

**Questions:**

Here are some of my quiestions.

1. It seems that there is a monotone property in your reward function, i.e., if we increase one arm's $x_n^T \theta_k$, the reward is increasing. Because of this, you can estimate the UCB and LCB by Eq. (2). So I am wondering what if we do not have such a monotone property? For example, in the case that each agent has its own score, and we want to maximize the average score (your case can be seen as the special one that the score is always 1). What can we do in this case? Also, are there any possible efficient approximate oracles in this case?

2. For the regret upper bound, we believe the factor of $T$ and $r$ are tight, then what about the other two factors? For the $K$ term, do we have a lower bound? Also, for the $\kappa$ term, there is a result that removes it in general linear model [1], could we use similar analysis to reduce it?

3. What's the exact value of $\alpha$ in your approximate version of the algorithm?


[1] Lee J, Yun S Y, Jun K S. A Unified Confidence Sequence for Generalized Linear Models, with Applications to Bandits[J]. arXiv preprint arXiv:2407.13977, 2024.



========After rebuttal=========

Thanks for the reply. I do not have further questions.

---

> ### Author Response · Authors · 2024-11-20
>
> We appreciate your effort in reviewing our paper and providing insightful and valuable feedback. Below, we have addressed each of your comments and questions.
>
> $\bullet$ **I think the primary contribution of this paper is proposing the new model setting. The contributions from the perspective of theoretical analysis seem limited, as they all follow the existing frameworks.**
>
> **Response:** Thank you for recognizing the novelty of our proposed model setting. While our theoretical contributions build on existing frameworks, we introduced significant extensions to address the unique challenges of our setting as follows. (Efficient Exploration of Combinatorial Matchings) Exploring all possible matchings naïvely is statistically expensive. We proposed a statistically efficient exploration strategy by constructing representative assortments $\{S\_{l,\tau}^{(n,k)}\}$ (Eq. 3), enabling effective exploration with reduced costs.  (Guarantee of Optimality) We also ensured that the optimal assortment remains in the active set $\mathcal{M}\_{\tau-1}$ through an inductive proof (Eqs. 5 and 6), a key step in achieving our regret bounds. (Use of G/D-Optimal Design) Lastly, we are the first to employ G/D-optimal design in matching bandits, enabling efficient learning of preferences through optimized exploration proportions $\pi\_{k,\tau}$.
>
> These contributions extend regret analysis to stochastic matching bandits under preference feedback with features, addressing high complexity from multi-agent and arm combinatorics. We believe these results establish valuable theoretical foundations for future research in similar settings.
>
>
>
> $\bullet$ **The SMB algorithm is not efficient, and its approximate version needs to use
> $\alpha$-oracle, but can only obtain an sublinear  $\alpha^2$-approxiamte regret upper bound.**
>
> **Response:** The computationally intensive part of the SMB algorithm is combinatorial optimization to determine assortments (Line 8 in Algorithm 1). However, due to the elimination-based design, there are only $O(\log(T))$ episodes, limiting the total number of updates to $O(\log(T))$.
>
> For the approximate version, the $\alpha$-oracle reduces the computational burden of exact combinatorial optimization. Although the regret scales with $\alpha^2$, this trade-off ensures scalability to larger problem sizes.
>
>
>
>
>
>
> $\bullet$ **Since the target and dynamics of the model is totally different with UCB-GS and ETC-GS, it is not really fair to compare with them.**
>
> **Response:**  We acknowledge that the use of baseline algorithms such as ETC-GS and UCB-GS may appear less directly relevant given their original design for a different objective. However, as mentioned in the experiment section, due to the novelty of our problem setting, there are currently no dedicated benchmark algorithms available for stochastic matching bandits under the MNL choice model with preference feedback.
>
> To ensure a meaningful comparison, we selected algorithms from the most closely related work in the matching bandits literature, such as ETC-GS and UCB-GS. While these methods were originally designed for deterministic arm behavior with known preferences, they provide a useful reference point to highlight the performance improvements and robustness of our proposed algorithm.
>
>
> $\bullet$ **Some minor typos:
> In Assumption 2, should it be $\inf_{\|\theta\|\le 1}$?
> In Eq. (4), it should be $R^{LCB}$**
>
>
> **Response:** Thank you for your comments and for pointing out the typo. We will revise Equation (4) to correctly use $R^{LCB}$.
>
> Regarding Assumption 2, we confirm that it is not a typo. The constraint $||\theta||\_2 \leq 2$ is intentional and arises from the properties of our estimator $\hat{\theta}\_{k,\tau}$, which is defined in $\mathbb{R}^r$. Specifically, as detailed in Line 1118, we show that $||\hat{\theta}\_{k,\tau}||\_2 \leq 2$ with high probability.
>
> We will clarify this in the final version to avoid any confusion. Thank you again for your careful review!

---

> > ### Author Response · Authors · 2024-11-20
> >
> > $\bullet$ **It seems that there is a monotone property in your reward function, i.e., if we increase one arm's $x_n^\top \theta_k$, the reward is increasing. Because of this, you can estimate the UCB and LCB by Eq. (2). So I am wondering what if we do not have such a monotone property? For example, in the case that each agent has its own score, and we want to maximize the average score (your case can be seen as the special one that the score is always 1). What can we do in this case? Also, are there any possible efficient approximate oracles in this case?**
> >
> >
> > **Response:** Thank you for your insightful question. Our current focus is on maximizing the number of successful matches, leveraging the monotonicity of the reward function. This allows us to use confidence bounds (UCB and LCB) effectively.
> >
> > For non-monotone cases, such as maximizing the average score where agents contribute distinct values,  a UCB-based approach may be better suited, as it can handle heterogeneous rewards without relying on monotonicity.
> >
> > We agree that extending our framework to such cases would be a potential avenue for future work.
> >
> >
> > $\bullet$ **For the regret upper bound, we believe the factor of $T$ and $r$  are tight, then what about the other two factors? For the $K$ term, do we have a lower bound? Also, for the $\kappa$ term, there is a result that removes it in general linear model [1], could we use similar analysis to reduce it?**
> >
> >  **Response:** Our regret bounds are **tight with respect to $T$ and $r$**, as they exhibit the expected $\sqrt{rT}$-style growth commonly observed in structured bandit literature. However, the dependency on $K\sqrt{K}$ in our regret bounds reflects the combinatorial complexity of matching multiple agents to arms with $K$ latent parameters. Since the regret accounts for successes across $K$ different arms with $K$ latent parameters, we conjecture that it is unlikely to improve beyond linear dependence on $K$. Achieving tighter regret bounds with respect to $K$ remains an open challenge.
> >
> > The dependency on $\kappa$ reflects the non-linearity of the model. The approach in [1] focuses on scenarios involving a single arm suggestion per round, whereas our framework involves multiple simultaneous choices, significantly complicating the analysis. Consequently, the methodology in [1] cannot be directly applied to our case. Addressing the dependency on
> > $\kappa$ in our setting is another open challenge that requires further investigation.
> >
> > $\bullet$ **What's the exact value of $\alpha$ in your approximate version of the algorithm?**
> >
> >  **Response:**  The value of $\alpha$ depends on the approximation policy. Using a simple greedy method, as described in Kapralov et al. (2013), we achieve
> > $\alpha$=1/2. This guarantees that the oracle provides at least 50\% of the optimal solution, ensuring the validity of the regret bounds while addressing computational challenges.

---

### Official Review · Reviewer_Vbm6 · 2024-11-03

**Soundness:** 3
**Presentation:** 3
**Contribution:** 2
**Rating:** 5
**Confidence:** 3

**Summary:**

This study introduces a stochastic matching bandit framework using the Multinomial Logit model with features. Agents are matched to arms, which stochastically choose agents or reject all. The goal is to minimize regret by maximizing successful matches.

**Strengths:**

1. The proposed model introduces an innovative framework for matching bandits, featuring novel stochastic behaviour of the arm that enhances its functionality in real-world application.
2. In addition to SMB (Algorithm 1), the integration of Algorithm 3 significantly addresses and mitigates computational challenges.

**Weaknesses:**

1. While the dynamic stochastic model presented is interesting, the objective in the given setting appears somewhat unreasonable. Specifically, in Section 3, the objective is to maximize the expected number of successful matches without regard to specific assignment matches, only focusing on whether there are no empty slots for each arm. This raises questions about the role of the MNL model introduced. It seems incongruent to assume the model includes preference feedback (in Line 172) when the objective disregards the resulting preferences, focusing solely on the absence/presence of assignment.

2. The significance of Theorem 1 and Theorem 3 is not clear. Notably, the dependency of $ K \sqrt{K} $ in the regret bounds is unusual within the bandit literature, where a dependency of $ \sqrt{K} $ is typically observed in most other bandit papers. Is  $ K \sqrt{K} $  unavoidable in this setting? Providing a tightness analysis for Theorem 1 and Theorem 3 would be valuable in elucidating the significance of these theoretical results.

3. The use of Baseline algorithms (i.e., ETC-GS and UCB-GS) is not suitable, as they are designed for a very different objective. Unfortunately, I don't have a good recommendation for the baseline.

**Questions:**

See Weaknesses.

---

> ### Author Response · Authors · 2024-11-20
>
> We appreciate your effort in reviewing our paper and providing insightful and valuable feedback. Below, we have addressed each of your comments and questions.
>
> $\bullet$ **While the dynamic stochastic model presented is interesting, the objective in the given setting appears somewhat unreasonable. Specifically, in Section 3, the objective is to maximize the expected number of successful matches without regard to specific assignment matches, only focusing on whether there are no empty slots for each arm. This raises questions about the role of the MNL model introduced. It seems incongruent to assume the model includes preference feedback (in Line 172) when the objective disregards the resulting preferences, focusing solely on the absence/presence of assignment.**
>
> **Response:** While our objective is centered on maximizing the successful match rate, the preference feedback generated by the MNL model plays a crucial role in the underlying learning process. The MNL model is not discarded when maximizing the number of successful matches; rather, it serves as a framework for learning the underlying preferences of arms (e.g., drivers, employers, etc.) over agents (e.g., riders, employees, etc.).  The probability of successful matching for an agent depends on the other agents assigned to the same arm based on the (relative) preference of the arm. \textit{Learning the preference utility for each agent to arm from preference feedback is crucial for maximizing successful matching in the system.} Without learning the preference utility, we cannot design an algorithm to maximize successful matching properly.
>
> To clarify, the success/failure of assignment (i.e., whether an arm successfully accepts at least one agent) in our objective is a higher-level goal, but the preference feedback generated by the MNL model is used for efficient exploration and learning about the arms’ true preferences over agents. This enables the algorithm to gradually refine the matching process and optimize long-term matching success by taking preference feedback into account.
>
>
> $\bullet$ **The significance of Theorem 1 and Theorem 3 is not clear. Notably, the dependency of $K\sqrt{K}$
>  in the regret bounds is unusual within the bandit literature, where a dependency of
>  is typically observed in most other bandit papers. Is $K\sqrt{K}$
>  unavoidable in this setting? Providing a tightness analysis for Theorem 1 and Theorem 3 would be valuable in elucidating the significance of these theoretical results.**
>
>
>
> **Response:** Below, we clarify the significance of our results while addressing the question of tightness.
>
> To the best of our knowledge, there are no existing methods tailored to our specific setting of stochastic matching bandits with MNL choice models and preference feedback. For comparison, we reference related work in the matching bandit literature under deterministic arm behavior with known preferences.  For example, the SOTA algorithm by [Kong \& Li (2023)] achieves a regret bound of $NK\log(T)/\Delta^2$, which translates to a worst-case regret of $\tilde{O}(NK^{1/3}T^{2/3})$ under  a specific $\Delta$. In contrast, our algorithm achieves a regret of $\tilde{O}(K\sqrt{KrT})$, which is fundamentally different in terms of scaling.
>
> Our regret bounds are **tight with respect to $T$ and $r$**, as they exhibit the expected $\sqrt{rT}$-style growth commonly observed in structured bandit literature. However, the dependency on $K\sqrt{K}$ in our regret bounds reflects the combinatorial complexity of matching multiple agents to arms with $K$ latent parameters. Since the regret accounts for successes across $K$ different arms at each time with $K$ latent parameters, we conjecture that achieving better than a linear dependence on $K$ in the regret is unlikely, implying that the regret lower bound lies between $K$ and $K\sqrt{K}$.
>
>
>
>  $\bullet$ **The use of Baseline algorithms (i.e., ETC-GS and UCB-GS) is not suitable, as they are designed for a very different objective. Unfortunately, I don't have a good recommendation for the baseline.**
>
> **Response:** We acknowledge that the use of baseline algorithms such as ETC-GS and UCB-GS may appear less directly relevant given their original design for a different objective. However, as mentioned in the experiment section, due to the novelty of our problem setting, there are currently no dedicated benchmark algorithms available for stochastic matching bandits under the MNL choice model with preference feedback.
>
> To ensure a meaningful comparison, we selected algorithms from the most closely related work in the matching bandits literature, such as ETC-GS and UCB-GS. While these methods were originally designed for deterministic arm behavior with known preferences, they provide a useful reference point to highlight the performance improvements and robustness of our proposed algorithm.

---

> > ### Author Response · Authors · 2024-11-29
> >
> > Thank you again for your valuable feedback and comments. We hope we have addressed your questions and concerns. As the deadline is approaching, please let us know if you have any further questions or comments.

---

> > > ### Comment · Reviewer_Vbm6 · 2024-12-02
> > >
> > > Thank you very much for your detailed response. I maintain my borderline score due to the issue regarding the dependency on $K\sqrt{K}$, as the authors have acknowledged the absence of a rigorous proof for the lower bound concerning $K$. However, regarding the overall quality of the paper, I would not object to its acceptance if other reviewers or AC strongly recommend the acceptance.

---

> ### Author Response · Authors · 2024-12-03
>
> Thank you for your comments. We would like to emphasize that a naive approach to eliminating suboptimal matching candidates results in a larger regret due to the $\Theta(K^N)$ possible matchings in this problem. In contrast, our regret bound significantly improves this to $O(K\sqrt{KT})$. Our algorithm utilizes a statistically efficient exploration strategy for each edge by constructing representative assortments $\{S\_{l,\tau}^{(n,k)}\}$ (Eq. 3), enabling effective exploration with reduced costs.

---

### Official Review · Reviewer_xYnY · 2024-11-04

**Soundness:** 3
**Presentation:** 2
**Contribution:** 3
**Rating:** 5
**Confidence:** 3

**Summary:**

This paper considers stochastic matching bandit that allows stochastic choice behaviors of arms and inclusion of outside options. Multinomial Logit (MNL) choice model with feature information is used to describe the unknown preferences of arms. The new bandit model has applications in ride-hailing services, online job markets and online labor markets. The goal is to maximize the likelihood of successful matches and learn the unknown arm preferences. The authors analyze the regret bound of an elimination-based algorithm. They show the regret bound achieves $\tilde{O}(K\sqrt{rKT})$ regret where r is the rank of feature space. The algorithm is shown to outperform two existing deterministic matching bandit algorithms via simulation experiments.

**Strengths:**

This paper fills an important gap in existing matching bandit literature that considers arm behavior as deterministic. It is indeed important in real world applications to not only focus on maximizing the likelihood of successful matches, but also learning arm preferences.
The paper develops an elimination-based algorithm to solve the stochastic matching bandit problem that can be divided into three phases, estimation, elimination and exploration. Detailed theory and proof is provided to prove the algorithm regret upper bound. The authors also discussed the computational complexity by optimization using $\alpha$-approximation oracle.

**Weaknesses:**

The important and practically relevant aspect of the setting is stochastic matching developed using MNL choice model with feature information. The powerfulness of the model is not demonstrated well. The experiment section assumes uniformly distributed features and does not cover the learning of arm performances. Theorem 1 is the key result for the paper. While detailed analysis is provided, it would be good to explain the key insights of the proof and clearly demonstrate how the three steps of the main stage contributes to the regret and which dominates the regret. Also, it will be good to have a full comparison of regret bounds and computational hardness with existing methods.

**Questions:**

• Can you further explain the $\pi$ in exploration step and how it affects the regret analyis?
• Tight regret bound is claimed for Thm 1. How does the regret bound compare to other matching bandit problem and is there a lower bound case?
• Outside option is considered in the paper. How about collision such that two arm choose the same agent?
• Please, see weaknesses and further elaborate on those points.

----------------------------After Rebuttal-----------------------------------
Thank you for the response. This helps me better understand the paper. G/D design seems to be the key - it would be good to revise Thm1 sketch proof. The current version is unnecessarily long.

---

> ### Author Response · Authors · 2024-11-20
>
> We appreciate your effort in reviewing our paper and providing insightful and valuable feedback. Below, we have addressed each of your comments and questions.
>
> $\bullet$ **The important and practically relevant aspect of the setting is stochastic matching developed using MNL choice model with feature information. The powerfulness of the model is not demonstrated well. The experiment section assumes uniformly distributed features and does not cover the learning of arm performances.**
>
> **Response:** Our choice of uniform distribution was driven by its simplicity and to provide a baseline evaluation of the proposed algorithm. We include an experiment with features drawn from a Gaussian distribution in Appendix A.5 in the revised paper. Importantly, the superior performance of our algorithm against the benchmarks holds under a variety of feature distributions, as the algorithm is designed to handle arbitrary feature information.
>
>
> We also note that the main objective is to maximize the number of successful matches in the overall system instead of a particular arm. This is why we measure the regret over the systems for our experiment rather than particular performance of arms.
>
>
> If the reviewer could suggest particular experiments, we would be happy to include additional experiments in the final version to further validate the robustness of our approach.
>
> $\bullet$ **Theorem 1 is the key result for the paper. While detailed analysis is provided, it would be good to explain the key insights of the proof and clearly demonstrate how the three steps of the main stage contribute to the regret and which dominates the regret.**
>
> **Response:** In our framework, regret arises solely from the exploration phase, where active edges corresponding to agents are explored to gather sufficient feedback for learning. The other steps of 'Estimation' and 'Elimination' do not directly incur regret but play a crucial role in managing exploration efficiently. Specifically, 'Estimation' ensures accurate parameter updates for the arms’ latent preferences, leveraging collected feedback to refine the confidence bounds used in subsequent episodes. 'Elimination' reduces the candidate set of active agents for each arm, excluding suboptimal edges to focus exploration on promising candidates. In the final version, we will add an explanation of these insights.
>
>
> $\bullet$ **Also, it will be good to have a full comparison of regret bounds.**
>
>
> **Response:**
>  To the best of our knowledge, there are no existing methods directly addressing the stochastic matching bandit problem under MNL choice with feature information as formulated in our paper. However, while it may not be fair to compare other regret bounds with ours, for reference, we provide a regret bound of an existing method in the matching bandits literature under deterministic arm behavior with known preferences.
>
> For example, the SOTA algorithm proposed in [Kong \& Li (2023)] achieves a regret bound of $NK\log(T)/\Delta^2$, where $\Delta$ is the minimum gap between preferences. This translates to a worst-case regret of $\tilde{O}(NK^{1/3}T^{2/3})$ for a fixed $\Delta$. In contrast, our algorithm achieves a regret bound of $\tilde{O}(K\sqrt{rKT})$. This highlights we have a tighter dependence on the time horizon $T$.
>
> We will include this discussion in our final version.

---

> ### Author Response · Authors · 2024-11-20
>
> $\bullet$ **Can you further explain the $\pi$ in exploration step and how it affects the regret analysis?**
>
>
> **Response:** As mentioned in lines 315–317, we utilize the G/D-optimal design framework [Lattimore \& Szepesvári, 2020] to determine the proportion $\pi\_{k,\tau}$ for exploring each edge (utility between an agent and an arm) in the active set $\mathcal{N}\_{k,\tau}$ during the exploration step. The purpose of this design is to ensure that we collect feedback efficiently to learn the optimal parameters $\theta\_k^*$ while minimizing unnecessary exploration.
>
> More specifically, the construction of $\pi\_{k,\tau}$ involves solving a G/D-optimal design problem. The solution assigns proportions to each element $n \in \mathcal{N}\_{k,\tau}$ such that the resulting sampling distribution optimally reduces uncertainty in the estimation of $\theta_k^*$ regarding active edges between agents and arms. This is achieved by finding an ellipsoid of minimal volume that contains all feature vectors $\{z\_n : n \in \mathcal{N}\_{k,\tau}\}$. Formally, the design ensures that the weighted $A$-norm satisfies:
> $$
> \max\_{n \in \mathcal{N}\_{k,\tau}} \|z\_n\|\_{V(\pi\_{k,\tau})^{-1}}^2 = r,
> $$
> where $V(\pi\_{k,\tau}) = \sum\_{n \in \mathcal{N}\_{k,\tau}} \pi\_{k,\tau}(n) z\_n z\_n^\top$. Here, $r$ represents the effective rank of the feature space, and the equality ensures a balanced exploration across the feature space.
>
> By leveraging the G/D-optimal design, $\pi_{k,\tau}$ ensures that we sample edges in a way that maximizes the information gain for estimating $\theta_k^*$. This prevents redundant exploration of already well-understood edges, which would unnecessarily increase regret.  The balanced exploration guaranteed by $\pi_{k,\tau}$ ensures that the contribution of the exploration step to regret diminishes over time. Specifically, the bound $\max\_{n \in \mathcal{N}\_{k,\tau}} \|z\_n\|\_{V(\pi\_{k,\tau})^{-1}}^2 = r$ allows us to derive a tight upper bound on regret by ensuring that the confidence intervals shrink uniformly across the feature space. This directly contributes to achieving the sublinear regret of $\tilde{O}(K \sqrt{rKT})$.
>
>
>  $\bullet$ **Is there a lower bound?**
>
> **Response:** Our regret bounds are *tight with respect to $T$ and $r$*, as they exhibit the expected $\sqrt{rT}$-style growth commonly observed in structured bandit literature. However, the dependency on $K\sqrt{K}$ in our regret bounds reflects the combinatorial complexity of matching multiple agents to arms with $K$ latent parameters. Since the regret accounts for successes across $K$ different arms with $K$ latent parameters, we conjecture that achieving better than a linear dependence on $K$ in the regret is unlikely. Achieving tighter regret bounds with respect to $K$ remains an open challenge.
>
>
>
>  $\bullet$ **Outside option is considered in the paper. How about collision such that two arm choose the same agent?**
>
>  **Response:** In our model, collisions are avoided by design due to the nature of the stochastic matching process.
>  Our model allows multiple agents to be assigned to the same arm in each time step, but each arm stochastically selects at most one agent from the assigned pool based on its preference, with the option to reject all assigned agents.
>
>  This mechanism mirrors real-world scenarios, such as ride-hailing services or online job markets, where arms (e.g., drivers or employers) choose at most one agent (e.g., a rider or employee) from their assigned pool at a given time.

---

> > ### Author Response · Authors · 2024-11-29
> >
> > Thank you again for your valuable feedback and comments. We hope we have addressed your questions and concerns. As the deadline is approaching, please let us know if you have any further questions or comments.

---

### Note · Authors · 2025-01-21

**Comment:**

Thank you for taking the time to review our paper.

**Withdrawal Confirmation:**

I have read and agree with the venue's withdrawal policy on behalf of myself and my co-authors.